# Continuous odor profile monitoring to study olfactory navigation in small animals

**Kevin S Chen[1†], Rui Wu[2†], Marc H Gershow[2,3]\*, Andrew M Leifer[1,4]\***

[1]Princeton Neuroscience Institute, Princeton University, Princeton, United States; [2]Department of Physics, New York University, New York, United States; [3]Center for Neural Science, New York University, New York, United States; [4]Department of Physics, Princeton University, Princeton, United States

**\*For correspondence:**
mhg4@nyu.edu (MHG);
leifer@princeton.edu (AML)

†These authors contributed equally to this work

**Competing interest:** The authors declare that no competing interests exist.

**Abstract** Olfactory navigation is observed across species and plays a crucial role in locating resources for survival. In the laboratory, understanding the behavioral strategies and neural circuits underlying odor-taxis requires a detailed understanding of the animal's sensory environment. For small model organisms like *Caenorhabditis elegans* and larval *Drosophila melanogaster*, controlling and measuring the odor environment experienced by the animal can be challenging, especially for airborne odors, which are subject to subtle effects from airflow, temperature variation, and from the odor's adhesion, adsorption, or reemission. Here, we present a method to control and measure airborne odor concentration in an arena compatible with an agar substrate. Our method allows continuous controlling and monitoring of the odor profile while imaging animal behavior. We construct stationary chemical landscapes in an odor flow chamber through spatially patterned odorized air. The odor concentration is measured with a spatially distributed array of digital gas sensors. Careful placement of the sensors allows the odor concentration across the arena to be continuously inferred in space and monitored through time. We use this approach to measure the odor concentration that each animal experiences as it undergoes chemotaxis behavior and report chemotaxis strategies for *C. elegans* and *D. melanogaster* larvae populations as they navigate spatial odor landscapes.

## Editor's evaluation

In this manuscript, the authors present a valuable tool and resource to create stable odorant landscapes (gradients) for the study of chemotaxis in small model organisms. Using a metal oxide-sensor-based approach, odor concentration is measured in space and time to produce data-driven simulations of the odor diffusion process. Combined with a system of distributed odorous air flows, odorant landscapes are generated to characterize aspects of olfactory navigation in *C. elegans* and the *Drosophila* larva.

## Introduction

Odor-guided navigation is common across the animal kingdom (*Baker et al., 2018*). Olfactory cues inform an animal of its location in a natural environment (*Boie et al., 2018*), and allow it to adjust its locomotion to navigate an odor landscape in a goal-directed manner (*Bargmann and Horvitz, 1991*; *Berg and Brown, 1972*; *Aceves-Piña and Quinn, 1979*). Odor-guided navigation is an ethologically relevant task that is important for the animal's survival, and it has been a useful framework with which to study genes and circuits underlying sensory-motor transformations (*Calhoun and Murthy, 2017*;

*Clark et al., 2013*). We focus in particular on the navigation of small model organisms in continuous, non-turbulent gradients established by the spread of odorants from their sources due to diffusion and drift. How animals interpret these gradients and use them to inform their actions remains an active and productive area of research especially in genetic model systems like *Caenorhabditis elegans* and *Drosophila melanogaster* (*Bargmann and Horvitz, 1991*; *Aceves-Piña and Quinn, 1979*; *Levy and Bargmann, 2020*; *Mattingly et al., 2021*; *Gomez-Marin et al., 2011*; *Gepner et al., 2015*; *Tastekin et al., 2018*).

A major challenge is to quantitatively relate the animal's behavior to the precise olfactory cue that the animal experiences moment-by-moment. Therefore, it is critical to precisely control the odor environment and record the sensory cues experienced by these animals. The need to *control* and *measure* odorants still pose a formidable challenge. While many techniques exist to present odor stimuli (*Gorur-Shandilya et al., 2019*; *Kadakia et al., 2022*), measure odor concentration (*Tariq et al., 2021*; *Celani et al., 2014*), or develop virtual odor environments in a lab (*Yamada et al., 2021*; *Radvansky and Dombeck, 2018*), it remains challenging for precise control and continuous monitoring of an odor landscape.

All approaches to generate odor landscapes in a lab environment must contend with the odor's diffusivity and interaction with other substrates. Early approaches to *control* odor concentration relied on passive diffusion to construct a quasi-stationary spatial odor gradient, for example by adding a droplet of odorant in a Petri dish in a 'droplet assay' (*Monte et al., 1989*; *Pierce-Shimomura et al., 1999*; *Louis et al., 2008*; *Iino and Yoshida, 2009*). Diffusion places severe limits on the space of possible landscapes that can be created and on the timescales over which they are stable, and the created odor profile is sensitive to adsorption of odor to surfaces, absorption into the substrate, temperature gradients, and air currents, all parameters that are difficult to measure, model, or control. Microfluidics allow water-soluble odors to be continuously delivered to a chamber in order to provide spatiotemporal control (*Chronis et al., 2007*; *Albrecht and Bargmann, 2011*; *Lockery et al., 2008*). Microfluidics devices, however, are limited in extent, require water-soluble odors, and must be tailor-designed to the specific attributes of the animal's size and locomotion. A microfluidic post-array has been shown to support *C. elegans* locomotion, but no similar microfluidic device has been demonstrated to support olfactory navigation of *Drosophila* larvae, for example.

We previously reported a macroscopic gas-based active flow cell that uses parallel flow paths to construct temporally stable odor profiles (*Gershow et al., 2012*). That approach allows for finer spatiotemporal control of the odor gradient, is compatible with *Drosophila* larvae, and works with volatile airborne-based odor cues. This device used an array of solenoid valves to generate programmable odor profiles, but perhaps because of its complexity has not been widely adopted.

Most methods to create an odor landscape do not provide a means for knowing or specifying the spatiotemporal odor concentration. In other words, while an experimenter may know that some regions of an area have higher odor concentrations, they cannot quantify the animal's behavior given a precise concentration of the odor. This limits the ability to quantitatively characterize sensorimotor processing. To address this shortcoming, various methods have been proposed to *measure* odor concentration across space. For example, gas samples at specific locations have been taken and measured offline (*Yamazoe-Umemoto et al., 2018*).

In one of the most comprehensive measurements to date, *Louis et al., 2008* and *Tadres et al., 2022*, used infrared spectroscopy to measure the spatial profile of a droplet-based odor gradient. They used this approach to study bilateral olfactory sensing and olfactory encoding in *Drosophila* larvae, clearly demonstrating the value of measuring odor landscapes. That work measured odor concentration at a relatively low temporal resolution ($\sim$ 1 sample/min), only when animals weren't present, and relied on a relatively expensive Fourier transform infrared spectroscopy machine.

In all of these previous works, measurements were performed offline, not during animal behavior, and the odor concentration was assumed to be the same across repeats of the same experiment, or when animals were present. But even a nominally stable odor landscape is subject to subtle but significant disruptions over time from small changes in airflow, from temperature variation, and from the odor's interaction with the substrate, which can include absorption, adhesion, and reemission (*Gorur-Shandilya et al., 2019*; *Yamazoe-Umemoto et al., 2018*; *Tanimoto et al., 2017*; *Yamazoe-Umemoto et al., 2015*). This is challenging to account for and control within a single behavior experiment, and is even more difficult to account for across multiple instances of such experiments. Additional

variability may also arise across experiments as a result of the introduction of animals, changes to agar substrates, and alteration in humidity or other environmental conditions. To recover the odor concentration that an animal experienced, there is a need to monitor odor concentration and animals' behaviors concurrently.

Our previously reported flow cell used a photo-ionization detector (PID) sensor moved across the lid before behavioral experiments to measure the odor concentration across space at a single point in time (*Gershow et al., 2012*). During experiments, the total concentration of odor in the chamber was monitored concurrently with measurements of behavior. While this provided some assurances that the overall odor concentration was relatively stable, it did not provide any spatial information concurrently with behavior measurements.

Metal-oxide sensors are small and scalable digital gas sensors that have recently been used to monitor odor concentration in studies of olfaction (*Schmuker et al., 2016*; *Drix and Schmuker, 2021*; *Tariq et al., 2021*) especially in the context of turbulent odor plumes. Arrays of metal-oxide sensors have been used to measure spatial odor landscapes by sampling across space (*Burgués et al., 2020*). Here, we leverage these approaches to present a new multi-sensor odor array with metal-oxide sensors combined with a flow chamber in order to measure the odor landscape that an animal crawls in with high spatial and temporal resolution. The array of sensors can be used in two ways: the full array can be used to measure the generated gradient throughout the extent of the chamber, or parts of the array can be used on the borders to monitor, *during behavioral experiments*, the odor profile along the boundaries of the chamber. By varying flow rates and the sites of odor introduction, we show a variety of odor landscapes can be generated and stabilized.

To demonstrate the utility of the apparatus, we applied this instrument to quantitatively characterize the sensorimotor transformation underlying navigational strategies used by *C. elegans* and *D. melanogaster* larva to climb up a butanone odor gradient. Butanone is a water-soluble odorant found naturally in food sources (*Worthy et al., 2018*) that is often used in odor-guided navigation studies and is of particular scientific interest for studies of olfactory learning (*Bargmann et al., 1993*; *Levy and Bargmann, 2020*; *Cho et al., 2016*; *Torayama et al., 2007*). We show that the agar gel used during behavioral experiments greatly disrupts an applied butanone gradient, and we demonstrate a pre-equilibration protocol allowing generation of stable gradients taking into consideration the effects of agar. Moreover, we monitor these gradients during ongoing behavior measurements via continuous measurements of the odor profile along the boundaries of the arena.

Using these stable and continuously measured butanone gradients, we measure odor-guided navigation in animals by tracking their posture and locomotion as they navigate the odor landscape. We record chemotaxis behavior and identify navigation strategies in response to the changing odor concentration they experience. In *C. elegans*, we observe navigation to the odorant butanone employs similar strategies to those reported for other sensory-guided navigation conditions, such as salt chemotaxis (*Iino and Yoshida, 2009*; *Dahlberg and Izquierdo, 2020*; *Luo et al., 2014*), namely a biased random walk strategy that adjusts teh frequency of turns, known as pirouettes (*Pierce-Shimomura et al., 1999*), and a gradual veering strategy, known as weathervaning (*Iino and Yoshida, 2009*; *Izquierdo et al., 2015*). We also measure navigation strategies of a mutant *C. elegans* and implicate the interneuron AIB mutant as being important for olfactory-guided weathervaning. In *D. melanogaster* larvae, we identify runs followed by directed turns (*Gershow et al., 2012*; *Louis et al., 2008*; *Gomez-Marin et al., 2011*). By using concurrent measurements of behavior and odor gradient, we characterize olfactory navigation in these small animals on agar with known butanone odor concentrations, which for *C. elegans* has not been reported before.

## Results

We developed new methods for both generating and measuring odor gradients which we describe here. The systems are modular, scalable, and flexible. The components, which can be used independently of each other, can be fabricated directly from provided files using online machining services.

### Flow chamber for generating spatiotemporal patterns of airborne odors

We first sought to develop a method of creating odor gradients that satisfied the following criteria:

1. The spatial odor profile should be *controllable*. Varying control parameters (e.g. flow rates, tubing connections) should result in predictable changes to the resulting odor landscape.
2. The odor profile should be *stable* and *verifiable*. The same spatial profile should be maintained over the course of an experiment lasting up to an hour, and this should be verifiable via concurrent measurements during behavior experiments.
3. The apparatus should be *straightforward* to construct and to use, and *flexible* to adapt to various experimental configurations, including using with either *C. elegans* or *Drosophila* larva, and with agar arenas of various sizes.

We constructed a flow chamber to control odor airflow across an arena (*Figure 1a*). Odor and humidified air are sourced from two bubblers, one containing pure water and the other an aqueous solution of odorant and water. Flow rates are controlled by separate mass flow controllers (MFCs) upstream of the bubblers. Downstream of the bubblers, the odor and air streams are divided into parallel sections of equal lengthed tubing. Each tube is connected to one input port of the flow chamber. The pattern of connections and the flow rates set by the two MFCs determines the shape of the produced odor profile. For instance, if odor is provided at a single central inlet, the resulting profile is a 'cone' (*Figure 1d–g*) whose peak concentration and divergence are controlled with the MFCs (e.g. speeding up the odor flow while slowing down the airflow broadens the cone). Temporal gradients can be achieved by varying the odor flow in time, subject to constraints imposed by the odor's absorption into the agar gel.

The outflux from the flow chamber is connected to a flow meter and a photo-ionization detector (PID) to monitor the overall flow rate and odor concentration respectively. The geometry of this flow chamber is shown in *Figure 1b*, where parallel tubings are connected from the side and the chamber is vacuum sealed with a piece of acrylic on top during experiments. The chamber is designed for use with ~100 mm square agar plates. The extra width (2.5 cm on either side) diminishes the influence of the chamber boundary on the odor profile over the arena. Interchangeable inserts allow for different agar substrates (e.g. circular plates) or for full calibration by odor sensor arrays (OSAs) (*Figure 1c*), discussed in the next section. Metal components are designed for low-cost fabrication by automated mechanisms (either laser cut-able or three-axis CNC machinable). The fabrication plans for the flow chamber and inserts (Data sharing), list of required components (*Table 1*, *Supplementary file 2*), and a detailed tutorial (*Supplementary file 1*) are all publicly available.

## Measuring the spatiotemporal odor distributions

A central difficulty in measuring animals' responses to olfactory cues is quantifying airborne odor concentrations that vary in space and time (*Vickers and Baker, 1994*; *Tariq et al., 2021*; *Kadakia et al., 2022*; *Yamazoe-Umemoto et al., 2018*). This difficulty is exacerbated in turbulent environments where odor plumes carry abrupt spatial and temporal jumps in concentration far from the source with fundamentally unpredictable dynamics (*Celani et al., 2014*; *Dennler et al., 2022*). But even in laminar flows, like those considered here, boundary conditions, slight changes in temperature, and the presence of absorbing substrates like agar make this challenging.

There is therefore a need, even for quasi-stationary gradients, to characterize the odor profiles in situ and to monitor these profiles during experiments. Various optical techniques, like laser-induced fluorescence or optical absorption (*Louis et al., 2008*; *Tadres et al., 2022*; *Demir et al., 2020*), exist to monitor concentration across planar arenas, but these have not been demonstrated for behavior experiments, are expensive to construct, require specially designed arenas, or some combination of these disadvantages. Electronic chemical sensors can reveal the time-varying concentration at a particular point in space. A tiled array of these sensors acts as a 'camera' forming a two-dimensional (2D) spatiotemporal reading of the concentration. The gold standard for measurement of odor concentration is the PID, but even the smallest versions of these sensors are both too large (~2 cm in all dimensions) and too expensive (~$500 each) to make an array. Metal-oxide odor sensors, designed to be used in commercial air quality sensors, are available in inexpensive and compact integrated circuit packages (*Schmuker et al., 2016*; *Burgués et al., 2020*). However, in general, commercial metal-oxide sensors are not designed for precision work - they tend to drift due to variations in heater temperature, humidity, adsorption of chemicals, and aging effects (*Dennler et al., 2022*). Most such sensors are designed to detect the presence of gas above a particular concentration but not to precisely measure the absolute concentration. We became aware of a newer metal-oxide sensor,

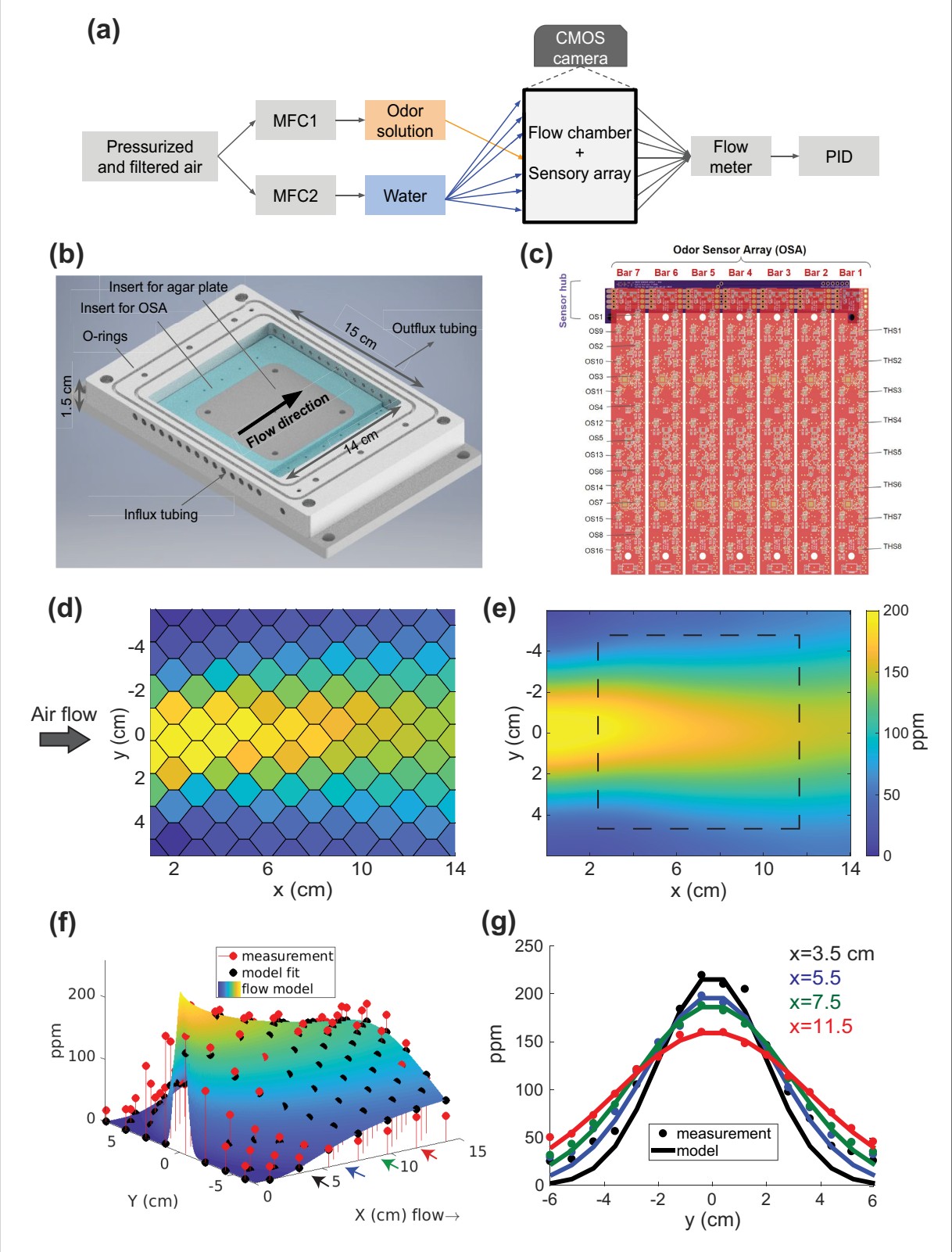

**Figure 1.** Odor flow chamber with controlled and measured odor concentration. (**a**) Schematic of airflow paths. Airflow paths for odor solution and water are controlled separately by mass flow controllers (MFCs) and spatially arranged into the odor chamber. The outflux from the chamber connects to a flow meter and photo-ionization detector (PID). (**b**) Flow chamber design. (**c**) Odor sensory array (OSA). Seven odor sensor (OS) bars are connected to a sensor hub. Each bar has 16 OS and 8 temperature/humidity sensors (THS). Measured odor concentrations from the OSA of a spatially patterned

*Figure 1 continued on next page*

*Figure 1 continued*

butanone odor concentration shown in (**d**) for each sensor, (**e**) interpolated across the arena with square dashed line indicating the area where agar and animals are placed. (**f**) A two-parameter analytic flow model fit to measurement. (**g**) Cross-sections from (**f**) at four different *x*-axis positions (shown as colored arrows). Sensor readouts are overlaid points on the smooth curves from a 1D diffusion model.

The online version of this article includes the following figure supplement(s) for figure 1:

**Figure supplement 1.** Long-term agreement between photo-ionization detector (PID) and metal-oxide sensors (odor sensor, OS).

**Figure supplement 2.** Temporal response of odor sensor (OS) to a step in odor concentration.

**Figure supplement 3.** Additional odor sensor controls.

**Figure supplement 4.** Characterization of temperature landscape.

**Figure supplement 5.** Odor profile is stable through time.

**Figure supplement 6.** Behavioral measurement is challenging in the presence of the full sensor array.

the Sensirion SGP30 that was designed for long-term stability and concentration measurement; we wondered if such a sensor could be calibrated for use in an OSA.

To calibrate the sensor, we created a controllable concentration source by bubbling air through butanone. The odor reservoir contains butanone dissolved in water and is kept below the saturation concentration (11 mM or 110 mM odor sources). We then mixed this odorized airflow into a carrier stream of pure air. We kept the carrier airflow rate constant ($\sim 400$ mL/min) and varied the flow rate through the odor source ($0 - 50$ mL/min); the odor flow rate was slow enough that the vapor remained saturated, so the concentration of butanone in the mixed stream was proportional to the flow rate through the butanone bubbler, as directly measured with a PID (*Figure 1—figure supplement 2*). We typically calibrated concentration with continuously ramped flow rate in triangle waves with a 500 s period for 2–3 cycles. We performed calibrations across a range of odor concentrations and against two PIDs, each with a different dynamic range (0–200 ppm, *Figure 1—figure supplement 1*, 0–1000 ppm, *Figure 1—figure supplement 3*), and also compared against the PIDs to characterize the metal-oxide sensors' temporal response (*Figure 1—figure supplement 2*). We found a one-to-one correspondence between the odor sensor reading and the PID reading that persisted over time and showed no obvious hysteresis. These metal-oxide sensors' temporal response and stability appear to be well matched for capturing the slow fluctuations in our stable and non-turbulent odor landscapes. We also characterized the heating effects of these sensors in the arena (*Figure 1—figure supplement 4*), and conducted measurements to show that powering the sensors does not noticeably change the concentration of odorant in the chamber (*Figure 1—figure supplement 3*). We reasoned that after calibrating the metal-oxide sensors to a PID, we could use the array to measure spatiotemporal odor concentration distributions.

We constructed a sensor array from 'odor sensor bars' (OSBs), printed circuit boards each containing 16 sensors in two staggered rows of 8. Each OSB also contained 8 temperature and humidity sensors

**Table 1.** Major components needed for assembling the device.
Detailed part's list is available in Supplementary File.

| Item | Manufacturer | Part number | Price estimate ($/unit) |
|---|---|---|---|
| Mass flow controller | Aalborg | GFCS-010654 | $900 |
| Photo-ionization detector | Ametek | piD-TECH | $500 |
| Metal-oxide sensors | Sensirion | SGP30 | $10 |
| PCB board | MacroFab and OSH Park | Customized | $400 |
| Flow chamber | eMachineShop | Customized | $250 |
| Flow meter | Aalborg | Model P flow tube meters | $160 |
| Teensy | PJRC | Teensy 4.0 | $25 |
| Camera | Basler | acA4112-30um | $3000 |
| Bubbler | Sigma-Aldrich | Duran GL 45 connection | $200 |
| Tubing connection | idex-upchurch | Fitting, manifold, and tubings | ~$1000 |

to allow compensation of the odor sensor readings. The OSBs are mounted orthogonal to the direction of airflow; 7 OSBs fit inside our flow chamber (112 sensors total) allowing a full measurement of the odor profile. Taken together these 112 sensors formed an OSA, capable of measuring odor concentrations with ~1 cm spatial resolution sampled at 1 Hz. Prior to all experiments we calibrated the OSA in situ by varying the butanone concentration across the entire anticipated range of measurement while simultaneously recording the odor sensor and PID readings.

To verify the ability of the OSA to measure concentration gradients, we created an artificially simple steady-state odor landscape and compared it to our predictions based on a convection-diffusion flow model. We flowed in odorized air ($\sim$ 30 mL/min) through a central tube and clean air through the others ($\sim$ 400 mL/min distributed into 14 surrounding tubings) in an environment without agar and without animals. This results in an airflow velocity $\sim$ 5 mm/s in the flow chamber. The odor concentration profile was established in a few minutes and remained stable over the timescale of 30 min behavioral experiments (*Figure 1—figure supplement 5*). As the flow rates and concentrations are all known and the flow is non-turbulent, and there is no agar or animals present, the concentration across the chamber should match a convection-diffusion model. After establishing the gradient, we recorded from the discrete odor sensors on the array (*Figure 1d*) and estimated the values in between sensors using spline interpolation with length scale equal to the inter-sensor distances (*Figure 1e*). We compared this stationary profile with a two-parameter convection-diffusion flow model (*Figure 1f and g*) fit to the data and described in the Materials and methods section. The measured concentrations in this artificially simplistic odor gradient show good agreement with the fit convection-diffusion model, especially in the central region where experiments are to be conducted, leading us to conclude that the OSA can accurately report the odor concentration. Note that the model serves as an independent

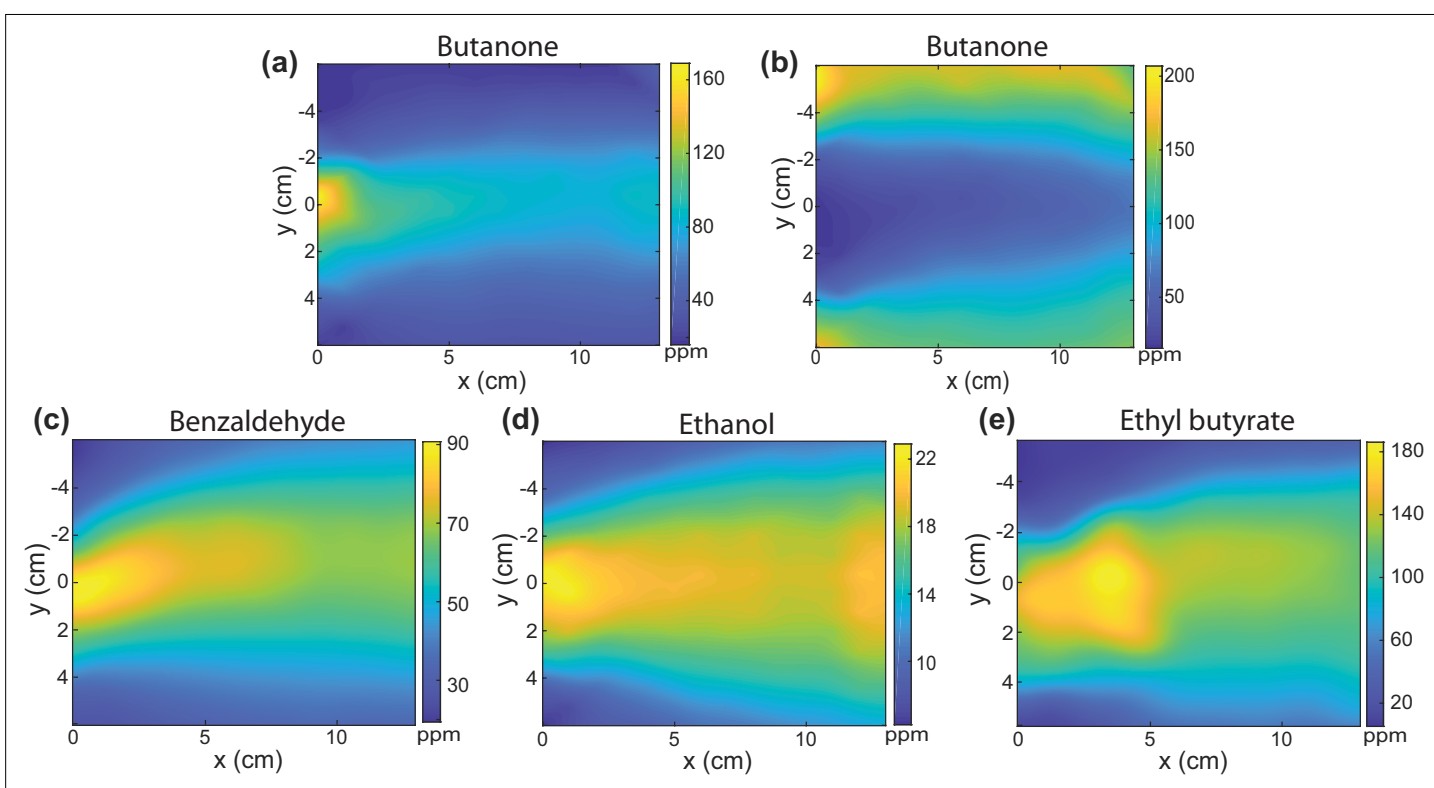

**Figure 2.** Flexible control of a steady-state odor landscape. (**a**) Configuration for a narrow cone with butanone. Colorbar shows interpolated odor concentration in ppm as measured by the odor sensor array. (**b**) An inverse cone landscape that has higher concentration of butanone on both sides and lower in the middle. Stationary cone-shape odor landscape of three different odorants: (**c**) benzaldehyde, (**d**) ethanol, and (**e**) ethyl butyrate.

The online version of this article includes the following source data and figure supplement(s) for figure 2:

**Source data 1.** Physical properties of odorants.

**Figure supplement 1.** Odor sensor (OS) calibration for different odorants.

validation of the sensors, and is not required to perform the measurement. We proceed to consider more complex odor environments.

We demonstrate several examples of flexible control over the odor profile. In the configuration shown in *Figure 1*, the center-most input provides odorized air and all surrounding inputs provide moisturized clean air to form a cone-shape stationary odor pattern. A narrower cone can be created by increasing the airflow (600 mL/min) of the surrounding inputs relative to the middle odorized flow (*Figure 2a*). The cone can be inverted by placing odorized air in the two most distal inputs, and clean air in all middle inputs (*Figure 2b*). This inverse cone has lower concentration in the middle and higher on the sides. In later animal experiments, we restrict odorized air to one side to form a biased-cone odor landscapes, resulting in a cone with an offset from the middle line of the arena. Many more configurations are possible, demonstrating that the odor flow chamber enables the flexible control of airborne odor landscapes that are much more complex than a single odor point source.

To demonstrate that the flow control and measurement methods are not restricted to any single odor molecule, we create and measure various stationary odor landscapes with benzaldehyde, ethanol, and ethyl butyrate odorants that, together with butanone, span a range of different vapor pressure and water solubility (*Figure 2*, *Figure 2—source data 1*). For each molecule, the metal-oxide odor sensors are separately calibrated against a PID (*Figure 2—figure supplement 1*). To our knowledge, most volatile organic compounds that can be sensed and calibrated by a PID should be compatible with our sensor array system. Depending on the vapor pressure, molecular weight of different molecules, and the geometry of our flow chamber, the cone-shape odor profiles can be skewed or less smooth. The skewed profile is possibly a result of the inhomogeneity in the pocket created by the odor sensor hub. Nonetheless, we empirically verified that these odor profiles are stationary and reproducible.

## Odor-agar interactions dominate classical butanone droplet assays

Classic chemotaxis experiments in small animals commonly construct odor environments with odor droplets in a Petri dish, usually with a substrate such as agar. Our odor delivery instrument is designed to be compatible with a similar environment. To first better understand classical chemotaxis experiments with butanone, we sought to characterize the spatiotemporal odor profile from an odor droplet point source using our OSA. We first considered the case without agar. In that case the odor concentration should be governed entirely by gas phase diffusion. We placed a 2 µL droplet of 10% butanone in water on the lid of our instrument centered in the arena above the full OSA and without any airflow (*Figure 3a and b*). Butanone was observed to diffuse across the arena in the first 3 min (Supplementary Video; *Figure 3—video 1*) resulting in an equilibrium concentration that was close to uniform across the odor sensors. We note that the final concentration of roughly 100 ppm and the equilibration timescale both match what we would expect from first principals for $\sim 10^{-6}$ mol of butanone in an $\sim 225\,\mathrm{mL}$ arena that have a diffusion rate of $\sim 0.08\,cm^2/s$ in air. A uniform odor landscape is not helpful for studying odor-guided navigation, but most behavioral experiments are not conducted in a bare flow chamber but contain a biologically compatible substrate, such as an agar gel, as is typically used in droplet assays. We therefore sought to investigate the role that agar plays in sculpting the odor landscape in butanone droplet experiments.

We introduced agar into the droplet assay by removing two sensor bars and replacing them with agar. We placed a butanone droplet directly on the agar or on the lid, as done classically in worm and larval assays respectively, and measured the odor landscape over time (*Figure 3b and c*). The odor concentration measured with agar is dramatically different from that measured without agar. Instead of quickly equilibrating to a uniform concentration, in the presence of agar there was instead a local maximum of butanone surrounding the droplet that persists even after 3 min. This difference in airborne odor concentration with and without agar persists after experimental perturbation such as removing and replacing the lid over the chamber (*Figure 3—figure supplement 1*). This convinced us of the need to account for odor-agar interactions when working with water-soluble odorants such as butane.

## Agar poses challenges for establishing stable odor landscapes

By placing the agar on the underside of the lid hanging above the odor sensors, we were able to simultaneously record spatial odor concentrations generated by our flow chamber opposite the surface of

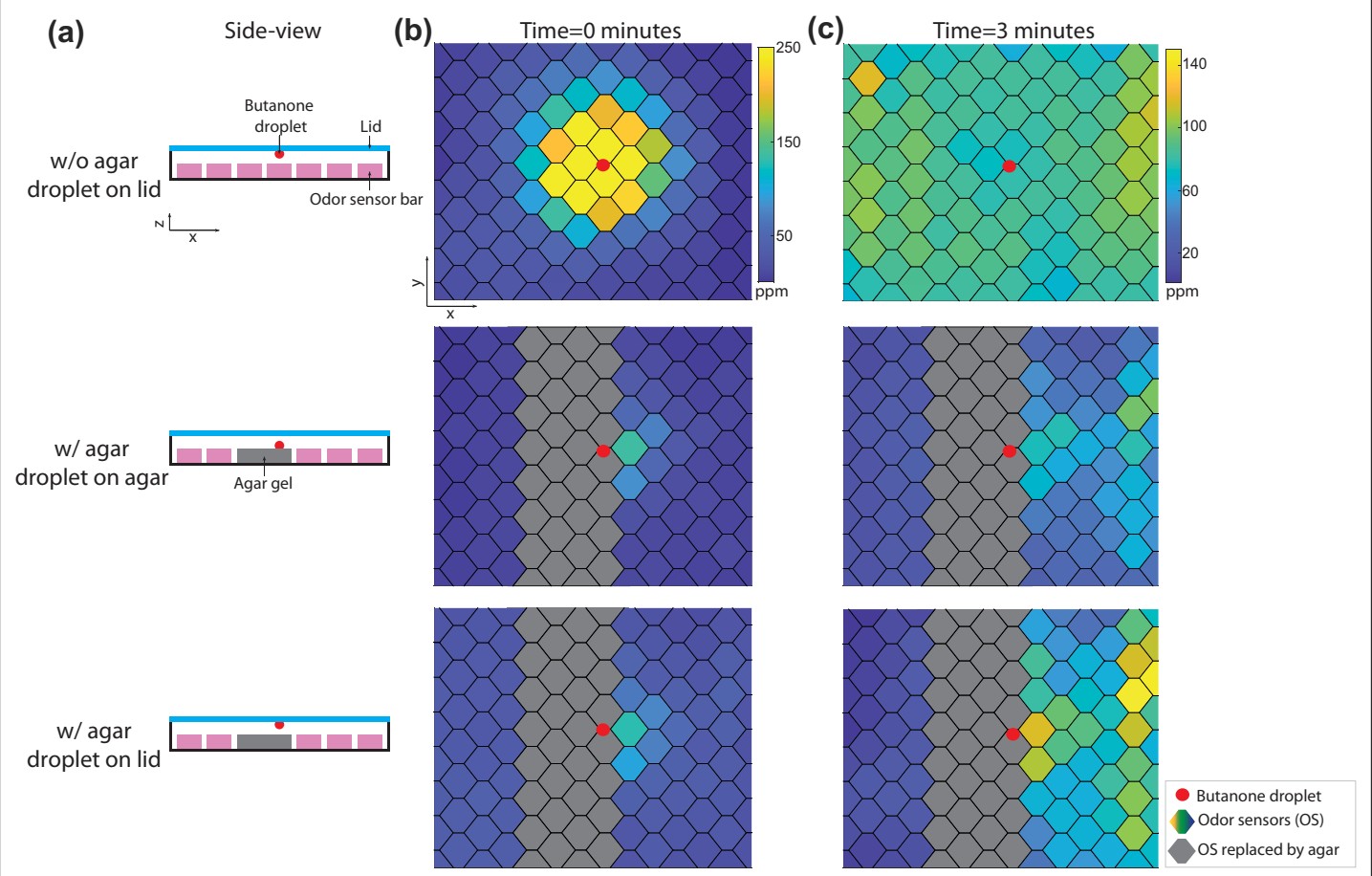

**Figure 3.** The presence of agar in the droplet assay alters the time evolution of butanone odor landscape. (**a**) Schematic showing configuration of an odor droplet in an arena with or without agar. Red dot indicates the position of butanone droplet (2 μL of 10% vol/vol butanone in water) on the lid (blue) or agar (gray). Odor sensor bars are shown in purple. (**b**) Spatial concentration measured by odor sensor array is reported immediately after butanone droplet is introduced into the arena. (**c**) Spatial concentration after 3 min.

The online version of this article includes the following video and figure supplement(s) for figure 3:

**Figure supplement 1.** Concentration measurements with an odor droplet and experimental perturbation.

**Figure 3—video 1.** Time evolution of odor landscape from a butanone droplet placed on agar (right) and without agar (left).

https://elifesciences.org/articles/85910/figures#fig3video1

the agar during animal behavior (*Figure 1—figure supplement 6*). Unfortunately, when imaging in this configuration, the odor sensors in the background obstruct the animals from being tracked over the extent of the arena. We therefore sought an alternative strategy: we chose to measure the odor profile along only the boundary of the arena in order to infer the odor concentration inside the arena in such a way as to also take into account the odor-agar interactions. This approach relies on establishing temporally stable odor landscapes, even in the presence of agar, which we sought to accomplish by leveraging our ability to control the flow of odor.

We first sought to measure how the presence of agar altered the odor profiles that would have been established by flow in a bare chamber lacking agar (*Figure 1*, *Figure 2*). As in *Figure 3c and d*, we replaced two OSBs with a rectangular strip of agar gel ('w/ agar') or a metal plate as a control ('w/o agar'), and then measured airborne odor concentration upstream and downstream of the agar under odorized airflow that would normally produce a cone profile (*Figure 4a–b*). While the agar had little effect on the odor landscape upstream of the agar, it drastically altered the downstream odor landscape (*Figure 4b*), suggesting that the agar absorbs the airborne butanone molecules. This finding is consistent with the odor droplet experiments (*Figure 3*), repeated with a full-sized 96 mm square agar plate intended for use with animals (*Figure 4c*), and to be expected since butanone is

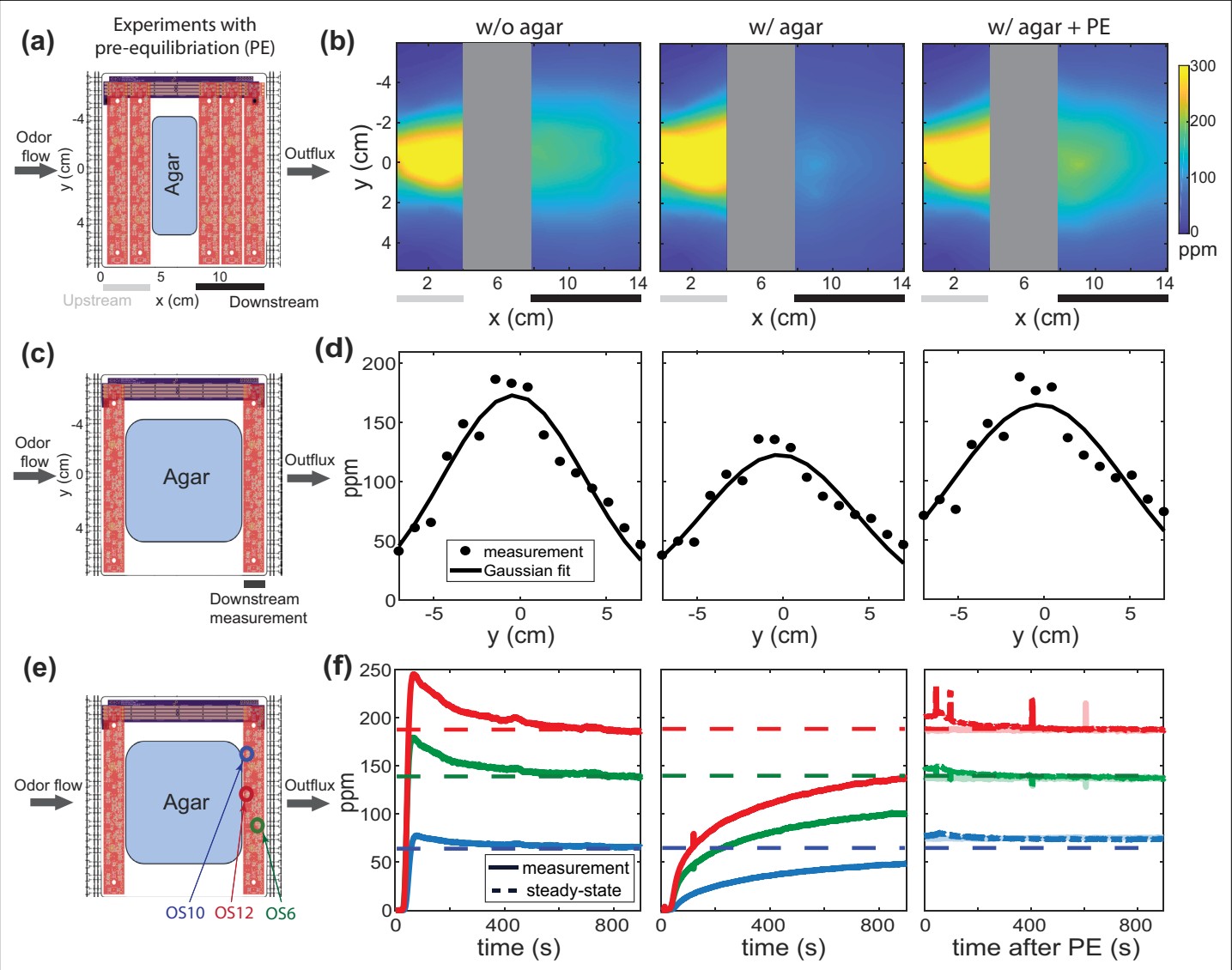

**Figure 4.** Under flow, agar interacts with butanone odor to disrupt the downstream spatial odor profile, but a pre-equilibration (PE) protocol can coax the system into quasi-equilibrium and restore a temporally stable odor profile. (a) Two odor sensor bars are replaced with agar to observe the effect of introducing agar on the downstream spatial odor profile. (b) Measured butanone odor profile is shown upstream and downstream of the removed odor sensor bars in the absence (left) and presence of agar (middle). Transiently delivering a specific higher odor concentration ahead of time via a PE protocol restores the downstream odor profile even in the presence of agar. (c) Additional odor sensor bars are removed and replaced with a larger agar, as is typical for animal experiments. (d) Measurements from the downstream sensor bar under the same three conditions in (b). The dots are sensor measurements and the smooth curve is a Gaussian fit. (e) The same experimental setup in (a), here focusing on time traces of only three downstream odor sensors (colored circles for selected OS). (f) Concentration time series of three sensors color-coded in (e). Traces for three conditions are shown: time aligned to initial flow without agar (left), time aligned to initial flow with agar (middle), and traces after PE (right, with transparent line showing measurements another 20 min after the protocol). The dash lines indicate the target steady-state concentration for each sensor.

The online version of this article includes the following figure supplement(s) for figure 4:

**Figure supplement 1.** Time series of concentration change that capture effects of agar gel and the pre-equilibration (PE) protocol.

highly soluble in water (275 g/L). Pulse-chase style experiments confirm that agar does indeed absorb and reemit butanone (*Figure 4—figure supplement 1b*). To accommodate the full-sized agar plate, we measured only the 1D odor profiles upstream and downstream of the agar (*Figure 4d*). Taken together, these experiments suggest that agar-butanone interaction presents a challenge for setting up and maintaining stable butanone odor landscapes.

## Odor flow-based compensation overcomes challenges posed by agar

We next sought a method to generate stable odor landscapes even in the presence of agar by applying a constant odor flow. In principle, the disruption caused by agar should be overcome by constant flow of a sufficiently long duration, after which the agar and airborne odor would be in quasi-equilibrium at all spatial locations, with the concentration of odor dissolved in the gel proportional to the airborne concentration above it. Howver, we measured odor concentration downstream of the agar and found that the airborne concentration failed to approach equilibrium on the timescales of single experiments (*Figure 4e and f*). This suggests that it is not practical to simply wait for the agar and odor to reach equilibrium. Instead, we developed a pre-equilibration protocol to more efficiently bring the system into equilibrium before our experiments.

To more rapidly establish a desired airborne odor landscape, we briefly first exposed the agar to an airflow pattern corresponding to higher-than-desired odor concentration, created by replacing the odor reservoir with one containing a higher concentration of butanone. We monitored the odor profile downstream of the agar until it reached the desired concentration and then switched to the original bubbler to maintain that concentration. Using this pre-equilibration protocol, we reached quasi-equilibrium quickly, typically after the order of 10 min (*Figure 4—figure supplement 1 and c*). Note the spatial parameters of the two airflow patterns were the same, only the concentration of the odor source changes. Pre-equilibration allows the generation of airborne odor gradients in the presence of agar that match those in the absence of agar (*Figure 4b, d, and f*, right vs left column), as indicated by odor sensor measurements along the boundary. This demonstrates that temporally stable gradients can be established even in the presence of agar.

## Monitoring the boundary determines the odor landscape at quasi-equilibrium

Pre-equilibration allows us to quickly establish a temporally stable odor landscape even in the presence of agar, and stability is indicated by monitoring the odor sensors along the boundary. Now we seek an approach to infer the odor concentration over the agar from the measurements along the boundary. In this section we will show that the odor concentration above the agar after pre-equilibration should be the same as the odor concentration when no agar is present and we will then validate this equivalence empirically (to within 10%). We will then use this knowledge to infer the odor concentration above the agar at quasi-equilibrium using the measurements along the boundary. If the airborne odor concentration along the boundaries upstream and downstream of the agar matches the profile in the absence of agar, one can infer that the airborne odor concentration landscape above the agar is also the same. We validated this approach in three ways: via analytical argument, via numeric simulation, and via direct empirical measurement.

The physical intuition is as follows: in the absence of sources or sinks, the fact that two odor concentration distributions obey the same differential equations and share the same conditions on all boundaries means that the distributions are identical throughout the interior. Agar adds complexity because it can act as a source or sink by absorbing and reemitting the odor, but even then, knowledge that the odor along the boundary is the same with and without agar places strong constraints on the interior odor landscape. The only way the interior could differ would be if sources and sinks of odor in the agar were precisely arranged so that all excess odor emitted from one point is exactly reabsorbed somewhere else before reaching the downstream boundary. Not only is such an arrangement unlikely, it is inherently temporally unstable and should tend to eliminate the excess sources and sinks over time. A mathematical version of this argument is presented in Appendix 1. Therefore, when we measure steady odor concentration along the boundary with agar that matches the concentration without, we are comfortable concluding that the odor landscape with agar is the same as without.

A numerical reaction-convection-diffusion model also shows that after pre-equilibration, the with-agar odor landscape matches the without-agar odor landscape (*Figure 5*). The model requires reasonable assumptions about the absorption rate, reemission rate, and capacity of the agar. The numerical simulation provides qualitative agreement to our empirical observations in *Figure 4*.

We experimentally verified that the airborne odor concentration above the agar matched our expectations based on measurements made along the boundary, by performing direct measurements of the airborne odor opposite the agar. We used an upside-down configuration as in *Figure 1—figure*

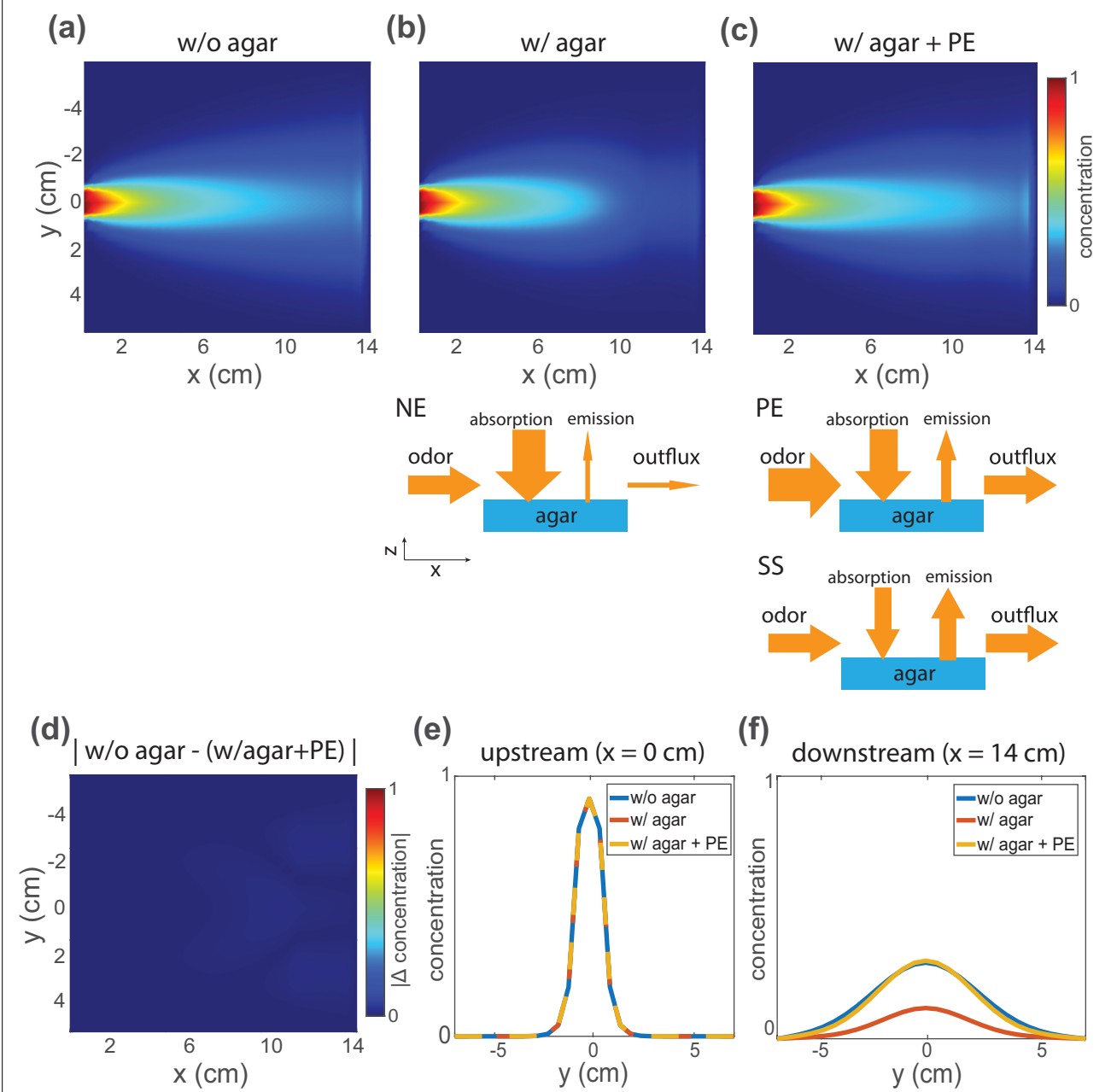

**Figure 5.** Simulations from a reaction-convection-diffusion model of odor-agar interaction show that at quasi-equilibrium the airborne odor concentration is the same with or without agar. (**a**) Simulation results of steady-state odor concentration in air without agar and with flow configured as in *Figure 1*. (**b**) The same simulation condition in (**a**) but now shortly after agar is introduced and before quasi-equilibrium is reached. A schematic of odor-agar interaction model is shown below. When agar is introduced, it absorbs the odor in air and decreases concentration measured downstream, producing a non-equilibrium (NE) concentration profile. (**c**) Odor concentration profile in air, with agar present, but after the pre-equilibration (PE) protocol brings this system to quasi-equilibrum. The PE protocol is shown in the schematic below, followed with steady state (SS) with the stable odor concentration profile shown above. (**d**) The absolute difference of concentration profile without agar (**a**) and with agar after PE (**c**) is shown. (**e**) Upstream and (**f**) downstream odor concentrations along the agar boundary are shown for all three conditions.

supplement 6 and *Figure 6*: we poured agar onto the lid and measured the odor concentration below it with the full sensor array underneath (*Figure 6*, *Figure 1—figure supplement 6*).

After the pre-equilibrium protocol, measured airborne concentration in the arena with agar was similar to measurements made without agar to within an average of less than 10% fractional difference in the area that the animal would crawl (*Figure 6b*). The agreement of odor landscape measured with

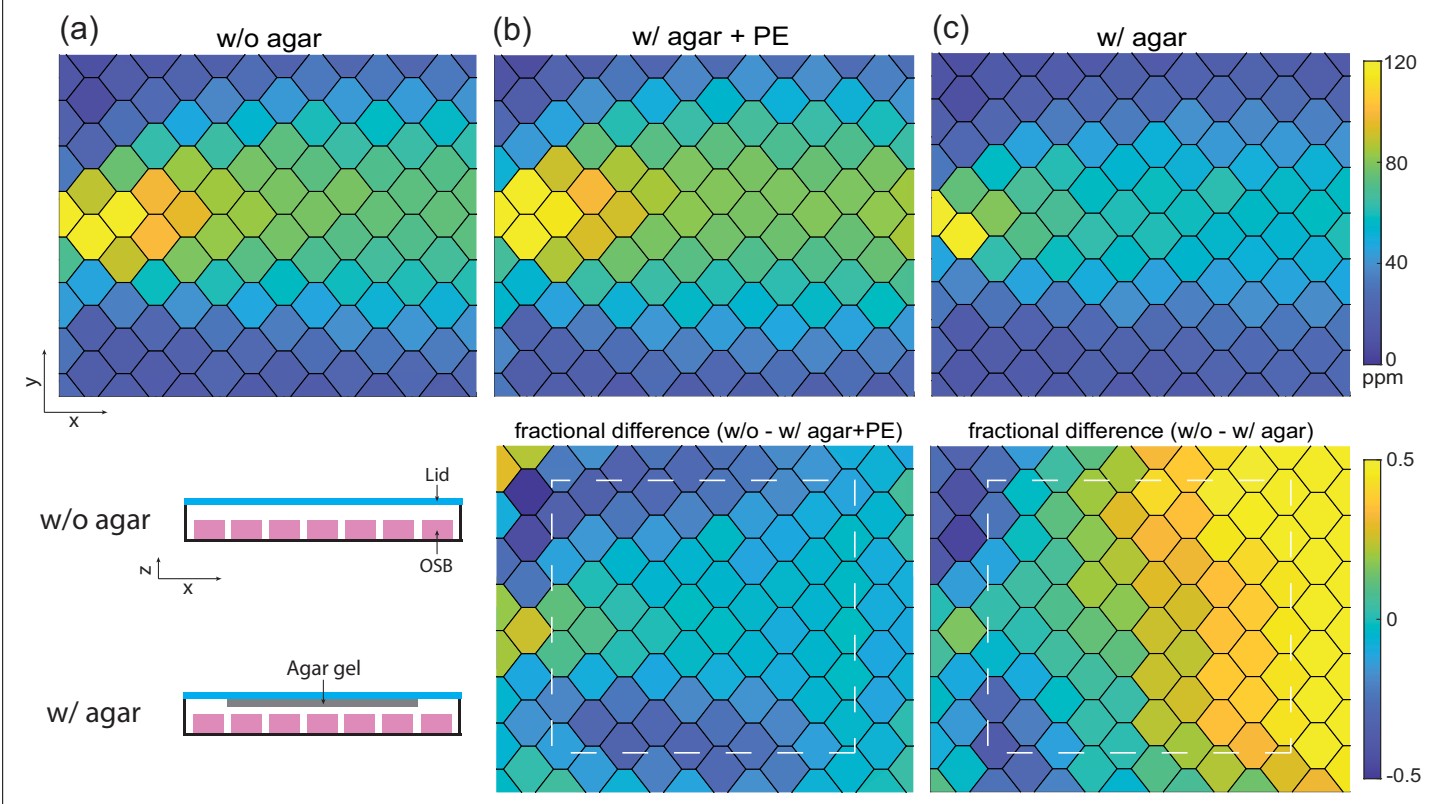

**Figure 6.** Direct measurement of odor concentration across agar confirms quasi-equilibrium odor profile. (**a**) Stationary butanone odor-cone profile without agar. Schematic for configuration with or without agar in this direct measurement shown below. (**b**) Same as (**a**) but with agar on the lid after the pre-equilibration (PE) protocol. Fractional difference between odor profile with and without agar is shown below and white dash line indicates the region with agar gel. (**c**) Same as (**b**) but without the PE protocol, 15 min into the recording. Fractional difference and agar gel region is shown below.

The online version of this article includes the following figure supplement(s) for figure 6:

**Figure supplement 1.** Quasi-equilibrium odor profile with agar is stable during animal experiments.

and without agar at quasi-equilibrium agrees with our modeling and indicates that the odor landscape is stable and identifiable in our setup by monitoring the odor profile along the boundary. Therefore, an experimenter could establish a desired odor gradient without agar, and then, even in the presence of agar, once the boundary measurements matched, the experimenter would be confident that the odor landscapes also matched. Note that our hypothetical experimenter would not be required to perform any simulations or calculations.

The presence of worms did not noticeably alter the odor concentration (*Figure 6—figure supplement 1*). Our approach relies on reaching quasi-steady state. In measurements made without the pre-equilibration protocol, and therefore presumably before quasi-steady state, the interior odor concentration with agar no longer matches the without agar, as expected (*Figure 6c*).

## Butanone chemotaxis in *C. elegans*

We sought to use our experimental approach to directly quantify *C. elegans*' navigation strategies for airborne butanone. *C. elegans* are known to climb gradients toward butanone (*Cho et al., 2016*; *Levy and Bargmann, 2020*). Microfluidic environments suggest that they use a biased random walk strategy to navigate in a liquid butanone environment (*Levy and Bargmann, 2020*; *Albrecht and Bargmann, 2011*). Worms are also known to use weathervaning to navigate airborne gradients of other odors (*Iino and Yoshida, 2009*; *Kunitomo et al., 2013*) although to our knowledge this has not been specifically investigated for butanone.

Worms were imaged crawling on agar in the flow chamber under an airborne butanone odor landscape illuminated by infrared light. Here, six recording assays were presented, with approximately 50-100 animals per assay, and two different odor landscapes were used. *C. elegans* navigated up the

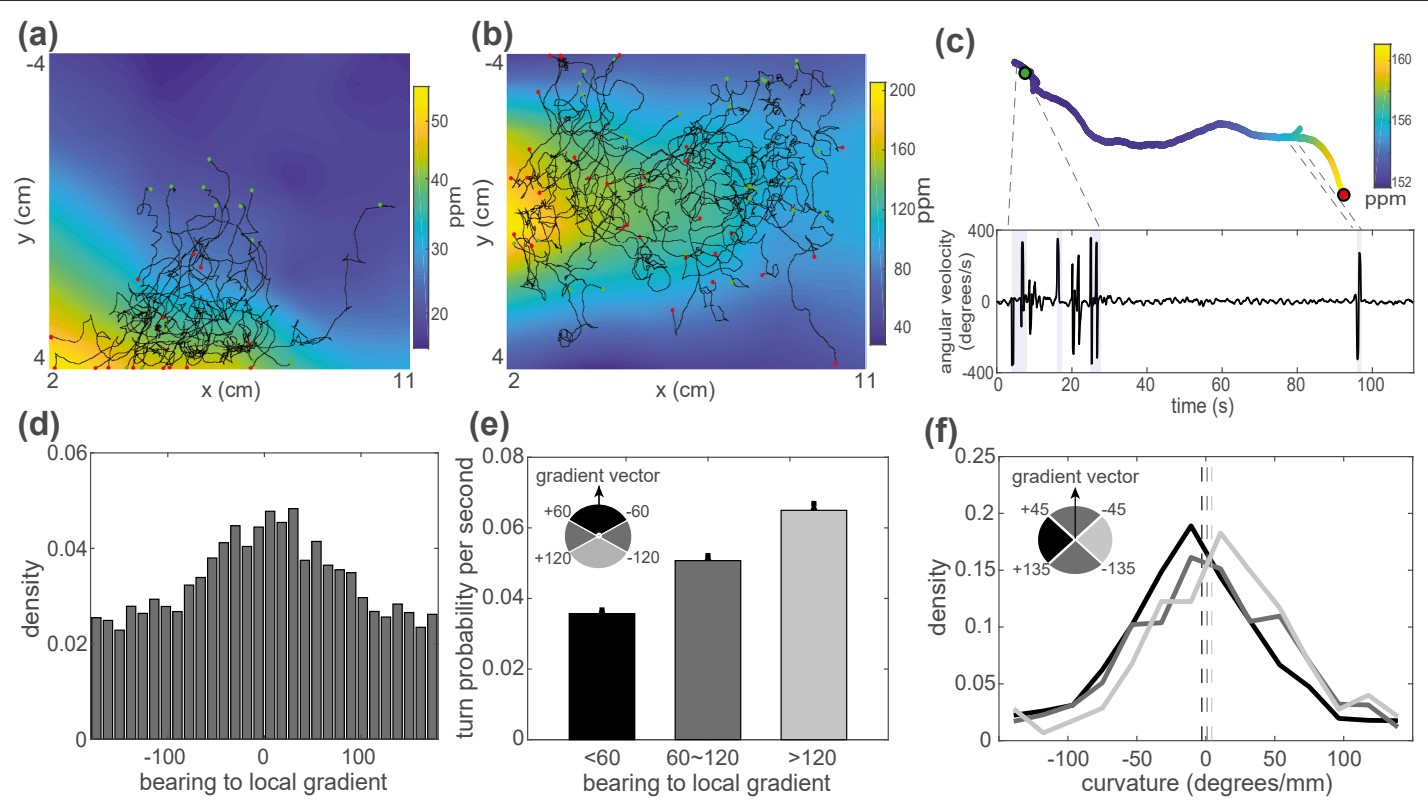

**Figure 7.** *C. elegans* use both biased random walk and weathervaning to navigate in a butanone odor landscape. Animals on agar were exposed to butanone in the flow chamber. (**a, b**) Measured animal trajectories are shown overlaid on airborne butanone concentration for different odor landscapes. Green dots are each animal's initial positions and red dots are the endpoints. (**c**) An animal's trajectory is shown colored by the butanone concentration it experiences at each position (top). Its turning behavior is quantified and plotted over time. Turning bouts are highlighted in gray. (**d**) Distribution of the animal's bearing with respect to the local airborne odor gradient is shown. Peak around zero is consistent with chemotaxis. (**e**) Probability of observing a sharp turn per time is shown as a function of the absolute value of the bearing relative to the local gradient. Modulation of turning is a signature of biased random walk. Error bars show error for counting statistics. Data analyzed from six experiments, ~100 worms per experiment, resulting in over 90 animal-hours of observation. (**f**) Probability density of the curvature of the animal's trajectory is shown conditioned on bearing with respect to the local gradient. Weathervaning strategy is evident by a skew in the distribution of trajectory curvature when the animal travels more perpendicular to the gradient. Bearing to local gradient defined by gray-scale quadrants. Three distributions are significantly different from each other according to two-sample Kolmogorov-Smirnov test (p < 0.001). Means are shown as vertical dashed lines.

The online version of this article includes the following figure supplement(s) for figure 7:

**Figure supplement 1.** Control measurements with airflow but no odor.

**Figure supplement 2.** Characterizing weathervaning with average curvature.

odor gradient toward higher concentrations of butanone, as expected (*Figure 7a and b*). Importantly, the odor concentration experienced by the animal at every point in time was inferred from concurrent measurements of the odor profile along the boundary of the agar (*Figure 7c*). On average, animals were more likely to travel in a direction up the local gradient than away from the local gradient, as expected for chemo-attraction (*Figure 7d*). We use the term 'bearing to local gradient' to describe the animal's direction of travel with respect to the local odor gradient that it experiences.

We find quantitative evidence that the worm exhibits both biased random walk and weathervaning strategies. To investigate biased random walks, we measured the animal's probability of turning (pirouette) depending on its bearing with respect to the local airborne butanone gradient (*Figure 7e*). We find that the animal is least likely to turn when it navigates up the local gradient and most likely to turn when it navigates down the gradient, a key signature of the biased random walk strategy (*Berg, 2018*; *Mattingly et al., 2021*).

To test for weathervaning, we measured how the curvature of the animal's trajectory depended on its bearing with respect to the local airborne butanone gradient (*Figure 7f*). When the animal's

navigation aligns to the axis of the butanone gradient, the distribution of the curvature of its trajectory was roughly symmetric and centered around 0 (straight line trajectory). By contrast, when the animal navigated more perpendicular to the gradient, the distribution of the curvature of its trajectories was skewed. The skew was such that it enriched for cases where the animal curved its trajectories toward the local gradient. We further calculate the average curvature as a function of bearing to local gradients following methods in *Iino and Yoshida, 2009*; *Figure 7—figure supplement 2a*. The result shows that the worm on average turns toward the gradient direction, which is the key signature of weathervaning strategy. Both the biased random walk and weathervaning behavior was absent in control experiments with flow but not odor, and we observed no evidence of anemotaxis at the ∼ 5 mm/s air velocities encountered by the animals (*Figure 7—figure supplement 1*, *Figure 7—figure supplement 2c*). We conclude that *C. elegans* utilize both biased random walk and weathervaning strategies to navigate butanone airborne odor landscapes (*Iino and Yoshida, 2009*; *Yoshida et al., 2012*). Both strategies had been observed for salt chemotaxis before, but to our knowledge had not previously been quantitatively characterized for airborne butanone. More broadly, previous measurements of navigation in response to airborne odors for *C. elegans* have not had access to the underlying odor concentration and therefore could only use proxies, such as the bearing angle or distance

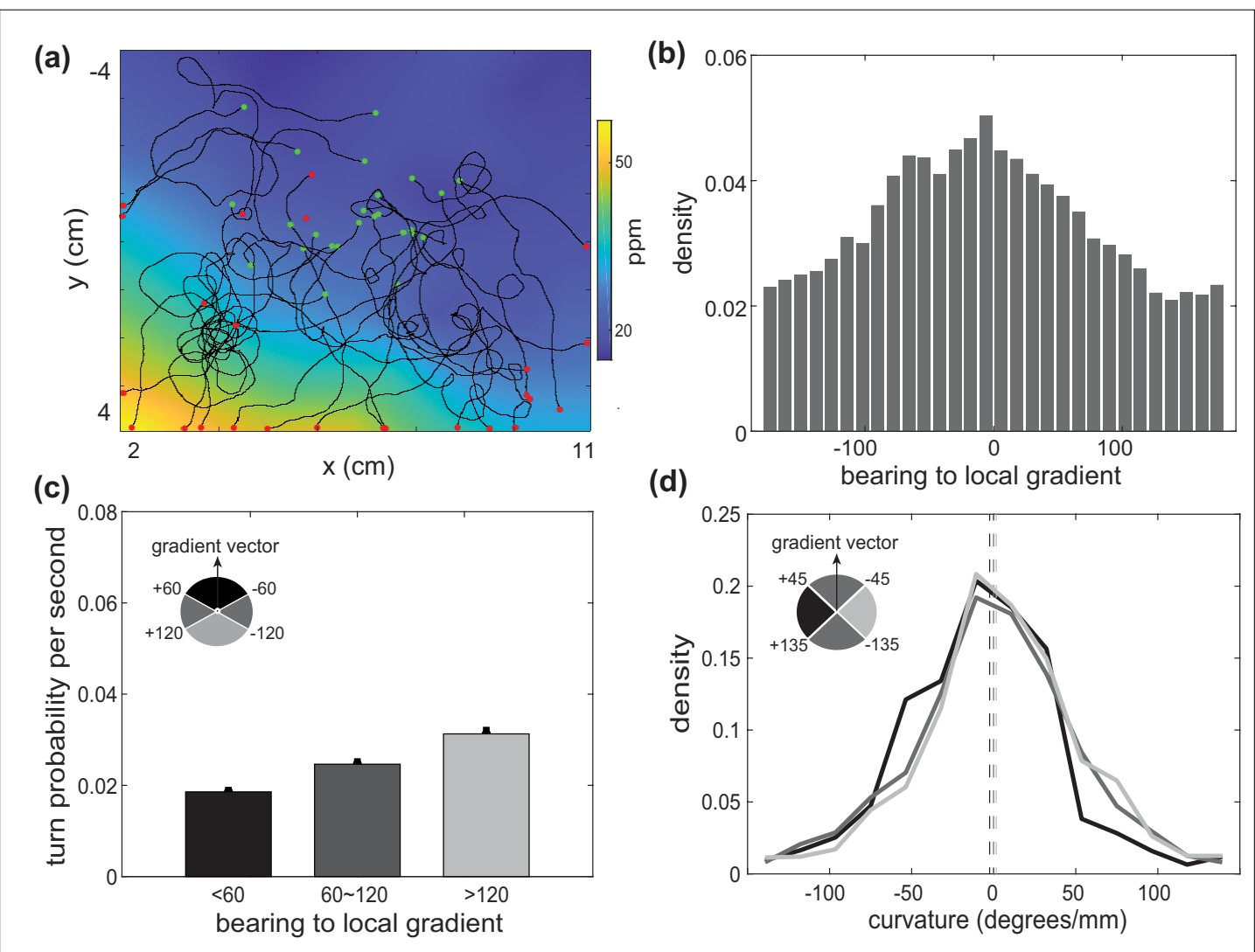

**Figure 8.** Butanone chemotaxis in worms with down-regulated AIB neuron. (**a**) Example behavioral trajectories from worms with down-regulated AIB neuron in the same butanone landscape. Green dots are each animal's initial positions and red dots are the endpoints. (**b**) Distribution of bearing to local gradients. (**c**) Turning probability at different bearing conditions. (**d**) Curvature conditioned on different bearing measurements. Data analyzed from three experiments, ~100 worms per experiment, resulting in over 51 animal-hours of observation.

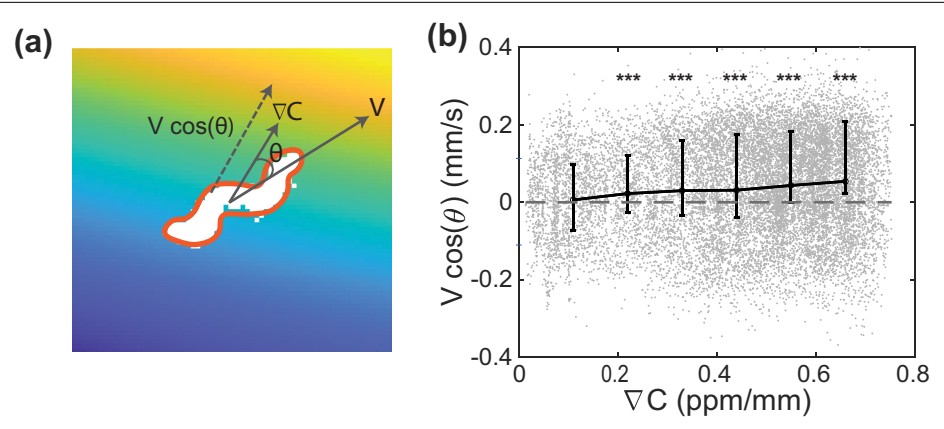

**Figure 9.** Chemotaxis performance as a function of gradient steepness. (**a**) Schematic of a worm tracked in the odor landscape. The crawling velocity vector $V$, local concentration gradient $\nabla C$, bearing angle $\theta$, and drift velocity $V\cos(\theta)$ are shown. (**b**) Tuning curve shows the drift velocity $V\cos(\theta)$ as a function of the odor concentration gradient. Gray dash line indicates an unbiased performance with zero drift velocity, gray dots are the discrete measurements, and the black line shows the average value within bins. Error bar shows lower and upper quartiles of the measurements. Other than the left-most bin with the lowest gradient, all other bins show significant difference from zero drift velocity using t-test (p<0.001, ***). Both two-sample t-test and Kolmogorov-Smirnov test (given the non-Gaussian distribution) show significant differences between bins and the first bin with the lowest concentration gradient (p<0.001).

with respect to a point odor source. Our approach allows a more direct description of the navigation strategy because we characterize turn probabilities and other features with respect to the local gradient directly (*Figure 7*), as opposed to a proxy.

To identify potential neurons involved in mediating the navigation strategy, we measured butanone-guided navigation of mutant animals in which the interneuron AIB was chronically inhibited. Disruptions to AIB had previously been shown to either increase weathervaning (AIB ablation) (*Iino and Yoshida, 2009*) or decrease biased random walks (*Luo et al., 2014*) (AIB inhibition) when navigating in a salt gradient. But to our knowledge, AIB's role in influencing the animal's butanone-guided navigation strategy had not previously been measured. We observed that AIB-inhibited animals still climbed butanone gradients, but the worm relied on different behavioral strategies (*Figure 8*). The turn probability was still biased by the concentration gradient suggesting that the animal still used a biased random walk strategy, although turning rate was overall lower. Most dramatically, the animal showed down-regulated weathervaning. In wild-type (N2) animals the distribution of the curvature of the animal's trajectory depends on its orientation to the local odor gradient (*Figure 7f*, *Figure 7—figure supplement 2a*) indicating weathervaning, but in AIB-inhibited animals the distribution of trajectory curvature is much less dependent on orientation to the local gradient (*Figure 8d*, *Figure 7—figure supplement 2b*). Together this suggests that AIB is important for supporting turning rate and the weathervaning strategy, and demonstrates the potential for this approach to characterize mutants or investigate neural circuits underlying navigation.

Having access to quantitative information about the odor the animal experiences provides an opportunity to characterize the animal's overall gradient climbing performance through observations of its local movements on a local gradient (*Mattingly et al., 2021*). We compute the animal's drift velocity as a function of local gradient steepness (*Figure 9*). This captures the animal's overall gradient climbing performance as a result of all the navigational strategies it uses, including the biased random walk and weathervaning. The analysis provides information relevant to a chemotaxis index, but it has the additional benefit of describing quantitatively how the animal responds to a local gradient and on what timescale. This analysis is only possible with a knowledge of the odor concentration experienced by the animal, and is naturally accessible in our instrument.

## Butanone chemotaxis in *Drosophila* larvae

To further evaluate the utility of the flow chamber for the study of small animal navigation, we investigated how larval *Drosophila* navigate butanone. Although butanone is not as commonly used as a

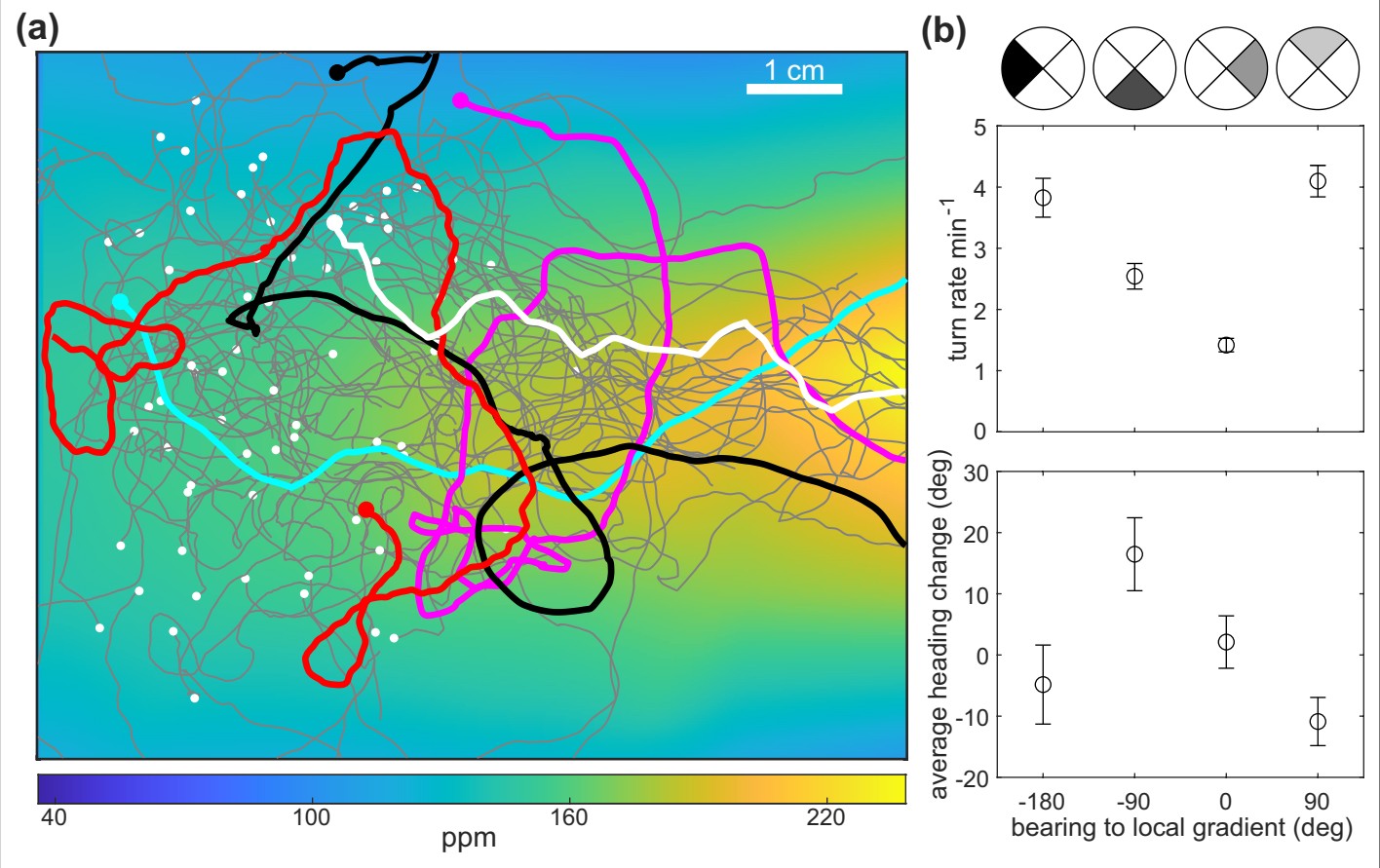

**Figure 10.** *D.melanogaster* larvae chemotaxis in the odor flow chamber. (**a**) Trajectories overlaid on the measured butanone odor concentration landscape. Example tracks are highlighted and the initial points are indicated with white dots. (**b**) Top: turn rate versus the bearing, which is the instantaneous heading relative to the gradient defined by quadrants shown on top. Error bar shows counting statistics. Bottom: average heading change versus the bearing prior to all turns (reorientation with at least one head cast). Error bars show standard error of the mean. Data analyzed from 6 experiments, 59 animals, with 620 turns resulting in over 6.8 animal-hours of observation.

stimulus with *Drosophila* as with *C. elegans*, butanone is known to be attractive to larval flies (*Dubin et al., 1995*; *Dubin et al., 1998*) and has been variously reported to be attractive (*Park et al., 2002*; *Jung et al., 2015*) and aversive (*Israel et al., 2022*; *Lerner et al., 2020*) to adult flies. To investigate the larva's navigational strategy in a butanone gradient, we created a 'cone'-shaped butanone gradient over the agar substrate using the pre-equilibration protocol, as before, and we confirmed the presence and stability of the gradient by continuously measuring the spatial distribution of butanone upstream and downstream of the agar arena. We monitored the orientation and movement of 59 larvae over six separate 10 min experiments (~ 10 larvae per experiment) with an average observation time of 7 min per larva (*Figure 10*).

Larvae moved toward higher concentration of butanone (*Figure 10a*). To analyze the strategy by which they achieved this, we first constructed a coordinate system in which 0° was in the direction of the odor gradient (toward higher concentration) and 180° was directly down-gradient; angles increased counterclockwise when viewed from above. We found that larvae initiated turns at a higher rate when headed down-gradient ($\pm 180°$ bearing with respect to the local gradient) than up-gradient (0°) (*Figure 10b*). When larvae turned, their reorientations tended to orient up-gradient (negative angle changes from $+90°$ bearing with respect to the local gradient, and positive angle changes from $-90°$) (*Figure 10c*). Bias in turning rate and reorientations in larvae are known navigation strategies in larvae (*Gershow et al., 2012*; *Gomez-Marin et al., 2011*; *Gomez-Marin and Louis, 2014*), but here it is to our knowledge the first characterization for butanone. With modulation of turning rate and orientation in the same odor environment, we conclude that *Drosophila* larvae use similar navigational

strategies to *C. elegans* to move toward butanone. We note that the chemotaxis efficiency is much higher in larvae compared to worms given the order of magnitude difference in run velocity (~3 mm/s in larvae and ~0.1 mm/s in worms).

## Discussion

We present a custom-designed flow chamber and OSA that enables us to measure navigation strategies of worms and fly larvae within the context of a controlled and measured odor environment. The key features of this odor delivery system are that (1) the odor concentration profile through space is controlled in the flow chamber, (2) the OSA provides a spatial readout to calibrate and measure the profile, and (3) the odor concentration profile is monitored during animal experiments. This last feature, the ability to monitor the spatial profile of odor concentration on the boundary during experiments, to our knowledge sets this method apart from previous approaches.

The odor landscape experienced by the animal can be recovered by one of two ways, with different tradeoffs. Typically, for a given set of flow parameters, we first quantify the odor gradient using the whole sensor array when agar is absent, then during behavioral experiments, we *confirm* the desired odor profile is achieved using sensor bars mounted along the boundary upstream and downstream of agar. During experiments with agar, the real-time monitoring of the odor profile at the edges of the arena guards against a range of possible errors or experimental imperfections. For example, exhaustion of the odor source, blocked connections, imperfect seals, and so on will all be detected by a change in the odor profile along the boundary, thereby avoiding silent changes to the odor landscape that may skew behavioral results. Indeed, it was this real-time monitoring that first alerted us to the fact that absorption of odor into the agar substrate caused the odor landscape to differ from our naive expectations.

If real-time monitoring of the entire arena during behavioral experiments is desired, the odor array can instead be mounted opposite the agar surface (*Figure 6*). We show that this introduces complications in imaging and makes tracking animals challenging (*Figure 1—figure supplement 6*). Future work could address these challenges through alternate lighting schemes (e.g. frustrated TIR in *Risse et al., 2013*) or by masking the sensor array with a darker material. For most applications, however, we expect that confirming the profile via sensors along the boundary will be sufficient because we have demonstrated that the odor landscape established by flow at quasi-equilibrium is stable and repeatable in the absence of experimenter or instrument errors (and detectable when not).

The ability to monitor spatial profile during experiments, along with a quantitative understanding of odor-agar interactions, provides confident knowledge of the odor experienced by the animal over time. This is especially important when we analyze time-varying navigation behavior in an odor environment that potentially changes in time too (*Yamazoe-Umemoto et al., 2018*; *Yoshida et al., 2012*; *Levy and Bargmann, 2020*). Past work focusing on neural circuit mechanisms for odor navigation in worms have often relied on coarser readout such as the chemotaxis index or using the distance to odor source as a proxy for the inverse of odor concentration. However, it has been shown that the exact odor concentration alters the animal's behavioral response to odor (*Yoshida et al., 2012*; *Levy and Bargmann, 2020*). Quantifying tuning curves with respect to more precise odor measurements may better constrain the investigations into neural mechanisms driving the sensorimotor transformations underlying navigation algorithms (*Clark et al., 2013*; *Liu et al., 2020*).

In contrast to liquid delivery of odor gradients via microfluidic chips (*Albrecht and Bargmann, 2011*; *Larsch et al., 2015*), our method allows worms to crawl freely on an agar surface. This allows our behavior measurements to be directly compared against classical chemotaxis assays (*Louis et al., 2008*; *Pierce-Shimomura et al., 1999*). Additionally, the macroscopic odor airflow chamber makes it straightforward to flexibly adjust the spatial pattern between experiments without the need to redesign the chamber. A detailed tutorial for assembling the instrument and the protocol for experiments is provided in *Supplementary file 1*.

Our setup uses low flow rates corresponding to low wind speed velocities (5 mm/s) to avoid anemotaxis. Larger organisms, including adult flies, navigate toward odor sources by combining odor and wind flow measurements (*Vergassola et al., 2007*; *Matheson et al., 2022*). Fly larvae exhibit negative anemotaxis at wind speeds 200–1000 times higher than those used here (*Jovanic et al., 2019*), but previous work showed that they do not exhibit anemotaxis at lower wind speeds like the ones used here (*Gershow et al., 2012*). For example, they do not exhibit anemotaxis at 12 mm/s which is still

higher than the velocities they experience here (*Gershow et al., 2012*). Therefore, we do not expect *Drosophila* larvae to exhibit anemotaxis under our flow conditions. *C. elegans* are not thought to respond to airflow. In experiments in aqueous microfluidic chips under flow, *C. elegans* move toward higher concentrations of attractant and do not respond to the flow of the liquid (*Albrecht and Bargmann, 2011*). In agreement, we do not observe any evidence of *C. elegans* anemotaxis in our chamber in response to control experiments without odor and with wind speeds of 5 mm/s (*Figure 7—figure supplement 1*).

We focused on the odor butanone because it is important for a prominent associative learning assay (*Torayama et al., 2007*; *Kauffman et al., 2011*). Butanone has high vapor pressure and is highly soluble in water and therefore it interacts strongly with agar. In this work we show that this odor-agar interaction makes it difficult to a priori infer an odor landscape experienced by the animal when agar is present, but that continuously monitoring the odor profile on the boundary overcomes this challenge.

Odorants with less solubility and lower vapor pressure may not require pre-equilibration to reach quasi-equilibrium quickly, but instead may be able to stabilize across agar on a shorter timescale. For example, ethyl butyrate is less water soluble (~0.586 g/100 mL in water, compared to 27.5 g/100 mL for butanone), has lower pressure (11.3 mmHg compared to 78 mmHg for butanone), and has a smaller gas phase diffusion coefficient ($5.6 \times 10^{-3}$ cm$^2$/s compared to ~69 cm$^2$/s for butanone). A droplet of ethyl butyrate has been shown to form a relatively stable odor landscape over a 5 min timescale in a dish on agar without flow (*Tadres et al., 2022*; *Louis et al., 2008*). Here, we demonstrate stable odor landscapes over longer timescales of at least 30 min with a more volatile odor, butanone, by using our flow-based assay (*Figure 1—figure supplement 5*). A strength of the flow-based approach with monitoring is that it should, in principle, allow for the use of a wider range of odors. Crucially, the continuous monitoring makes this approach more robust against other unforeseen odor interactions, such as interactions between an odorant and glass, aluminum or plastic.

Here, we have addressed the problem of creating non-turbulent airborne odor landscapes. The biophysical processes governing odor sensing in small animals such as *C. elegans* are not fully understood. The worm carries a thin layer of moisture around its body as it moves on the agar substrate (*Bargmann, 2006*) and it is unclear to what extent the worm pays attention to the concentration of an odorant in the agar below it vs the air above it. Our reaction-convection-diffusion model suggests that at the quasi-equilibrium conditions used in our experiments the odor concentration in agar is related to the airborne odor concentration directly above it up to a scalar that we predict to be constant across the agar. Although we have not measured this empirically, this suggests that even in the extreme case that the animal only senses odor molecules in the agar, the odor concentration experienced by the animal in our experiments should differ by no more than a scaling factor compared to our estimates based on the airborne odor concentration.

Here, we also showed the possibility of and challenges associated with directly measuring the full spatial odor profile (instead of using the boundary) during simultaneous behavior tracking (*Figure 1—figure supplement 6*). Future work is needed to improve concurrent animal tracking methods, to design more miniaturized and scalable sensors, and to generalize to more complex or 3D environments.

Knowing the concentration experienced by the animal is not only useful for measuring navigational strategies more precisely than in classical assays, like the droplet chemotaxis assays. It will also be crucial for studying *changes* in navigational strategy, such as those in the context of associative learning (*Cho et al., 2016*; *Torayama et al., 2007*), sensory adaptation (*Levy and Bargmann, 2020*; *Itskovits et al., 2018*), and long timescale behavioral states (*Calhoun et al., 2014*; *Gomez-Marin et al., 2011*; *Klein et al., 2017*). In all those cases, it will be critical to disambiguate slight changes to the odor landscape from gradual changes in the navigational strategies. Continuously monitoring the odor landscape during behavior will remove this ambiguity.

## Materials and methods

**Key resources table**

| Reagent type (species) or resource | Designation | Source or reference | Identifiers | Additional information |
|---|---|---|---|---|
| Strain, strain background (*C. elegans*) | N2 | CGC | RRID:WB-STRAIN:N2 | |
| Strain, strain background (*C. elegans*) | AML580 | This paper | | |

*Continued on next page*

*Continued*

| Reagent type (species) or resource | Designation | Source or reference | Identifiers | Additional information |
|---|---|---|---|---|
| Strain, strain background (*C. elegans*) | ZC2406 | *Luo et al., 2014* | | Gift of Yun Zhang |
| Genetic reagent (*D. melanogaster*) | NM91 | *Coen et al., 2014* | | Gift of Mala Murthy |

## Odor flow chamber

### Flow chamber setup

The odor chamber (*Figure 1b*) was machined from aluminum (CAD file in supplementary Data sharing section). The chamber is vacuum sealed with an acrylic lid. The inner arena contains an aluminum insert that can hold the odor sensor array or a square Petri dish lid (96×96 mm$^2$). The heading in which airflow can travel above the insert in the arena is 1 cm tall. The whole setup is mounted on an optical breadboard and enclosed in a black box during imaging.

The airflow system is connected to a pressurized air source, passing through a particulate filter (Wilkerson F08) and a coalescing filter (Wilkerson M03), then regulated by MFCs (Aalborg GFC). MFCs are controlled via a Labjack D/A board from a computer using custom Labview code. We modulate the flow rate bubbling through liquid in enclosed bottles (Duran GL 45). The moisturized or odorized air is then passed into the flow chamber through inlet tubings. The outlets are connected to a copper manifold, then passed to a flow meter to assure that the inlet and outlet flow rates match. An optical flow sensor is fixed on the flow meter to make time stamps for opening and closing of the lid of the flow chamber during animal experiments. A PID (piD-TECH 10.6 eV lamp) is connected to the outlet of airflow, providing calibration for the OSA and detection of air leaks or odor residuals in the system. Output readings from the PID, MFC, odor sensors described in the next section, and imaging camera, are all captured on the same computer sharing the same clock. Analog signals from the PID readout and MFC readback are digitized via a Labjack and recorded with the Labview program.

## Odor flow control

To construct different odor landscapes tubes from the liquid-odor and water reservoirs are connected to the flow chamber in different configurations. For a centered 'cone-shape' odor landscape the tubing carrying odorized airflow is connected to the middle inlet. For the 'biased-cone' landscape, the tubing for odorized air is connected to the inlet 4 cm off-center. For uniform patterns, all are connected to the same source through a manifold. For all experiments the background airflow that carries moisturized clean air is set to ~400 mL/min, except for *Figure 2* where this value was varied. The odor reservoir contains either a 11 mM or 110 mM butanone solution in water with ~30 mL/min airflow bubbling through the liquid. For other molecules, we prepared odor reservoir with benzaldehyde (1/250 vol/vol in water), ethanol (1/10 vol/vol in water), and ethyl butyrate (1/150 vol/vol in water).

Overall flow rates across the chamber in experiments were always around or less than ~400 mL/min to avoid turbulence. We confirmed that this regime had no turbulence by visualizing flow in a prototype chamber using dry ice and dark field illumination. Our empirical observations matched theory: Given that the chamber is 15 cm wide and 1 cm deep, a flow rate up to 1 L/min corresponds to ~1.1 cm/s. With kinematic viscosity of air $\sim 0.15$ cm$^2$/s, the Reynolds number is 7.3 times the flow rate in L/min, which is below the turbulence onset (Re = 2000).

## Odor sensor array

A spatial array of metal-oxide based gas sensors (Sensorion, SGP30) along with a relative humidity and temperature sensors (ams, ENS210) was used to measure the odor concentration field in the flow chamber. Sensors are arranged together into groups of 16 odor sensors and 8 humidity sensors on a custom circuit board (MicroFab, Plano, TX, USA) called an OSB. The 16 odor sensors are arranged in two interleaved columns spaced 1 cm apart. The vertical distance between sensors in each column is 1.5 cm, which results in 0.75 cm vertical sensor resolution. OSBs are in turn plugged into a second circuit board (OSH Park, Portland, OR, USA) called the odor sensor hub. OSBs can be added or removed in different arrangements depending on the experiment, for example to make room for agar. The distance between sensor columns across neighbor OSBs is 1 cm. Depending on the experiment,

up to 112 odor sensors are arranged in a triangular grid such that no sensor directly blocks the flow from its downstream neighbor, accompanied by 56 humidity sensors in a rectangular grid.

Sensors are read out via the I2C protocol. Each SGP30 sensor has the same I2C address, as does each ENS210 sensor (different from the SGP30); to address multiple sensors of the same type we use an I2C bus multiplexer (NXP, PCA9547). Each OSB contains two multiplexers for its 16 sensors. The multiplexers are also addressed over I2C and can have one of eight addresses (three address bits). On each board, the two multiplexers share two bits (set by DIP switches); the remaining bit is hardwired to be opposite on the two multiplexers. Thus, each OSB can have one of four addresses set by DIP switches, and four OSBs can be shared on one I2C bus.

To communicate with the sensors, we used a Teensy 4.0 microcontroller (PJRC, Sherwood, OR, USA) running custom Arduino software. While the Teensy has two I2C busses, we found it more straightforward to use two microcontrollers instead. Both microcontrollers communicated via USB serial to a desktop computer running custom LabView software. Measurements from all sensors are saved to computer disk in real time. Readouts from the humidity sensors are also sent to their neighboring odor sensors in real time for an on-chip humidity compensation algorithm.

## Heat management

To avoid generating thermal gradients, the system has been designed to dissipate heat to the optics table. Each metal-oxide odor sensor contains a micro hotplate which consumes 86 mW power during readings. To dissipate this heat the aluminum insert inside the flow chamber serves as a heat sink. OSBs are connected to the insert using heat conductive tape and thermal paste. The insert and chamber are in turn in direct thermal contact with the optics table. Temperature and humidity is constantly monitored at eight locations per OSB via the on-board temperature and humidity sensors during experiments to confirm that there is minimal thermal or moisture gradient created in the environment.

We experimentally validate heat management when the full sensor array is active for odor measurements. The temperature increases in the first 5 min of recording and later stabilizes (*Figure 1—figure supplement 4*). After stabilizing, the temperature readout shows modest spatial fluctuations on the scale of 0.5°C. In separate experiments with agar, we measured similar spatial fluctuations on the agar surface. Furthermore, we note that the control experiment with airflow and active sensors but without odor molecules show little thermotaxis or biased trajectories, therefore concluding that the heat effects are minor for animal experiments.

## Measurements and calibration

We measure from the odor sensors at 1 Hz for both calibration and behavior experiment modes. We sample from the PID at up to 13 Hz. We synchronize and time align the measurements from the OSA, MFC read-back, and PID recording with the same computer clock.

To calibrate the odor sensors to the PID as in *Figure 1—figure supplement 2*, a spatial uniform flow was delivered in a triangle wave or a step pattern. Time series from each odor sensor and the downstream PID were aligned by time shifting according to the peak location found via cross-correlation. The time shift was confirmed to be reasonable based on first principle estimates form the flow rate.

After measuring odor sensors' baseline response under clean moisturized air for 5 min, an odorized air was delivered. To fit calibration curves, the raw sensor readout was fit to the PID measurements with an exponential of form:

$$\text{PID}(t) = A \exp(B * \text{OS}(t - \tau)) \tag{1}$$

where $\text{PID}(t)$ voltage is on the left-hand side, the scale factor $A$ and sensitivity $B$ are fitted to match the raw sensor reading $\text{OS}(t - \tau)$ that is time shifted by time window $\tau$. This fitted curve maps from raw readings to odor concentration for each sensor. We validate the fitted curve across different recordings. The distribution of the coefficients $A$ and $B$ are relatively uniform across sensors in the middle of the arena. The sensor mapping are also reliable, so using ±std of the fitted curve changes less than 10% of the overall concentration scale of the landscape.

## Models for odor flow and odor-agar interaction

We use two models in our work: (1) a convection-diffusion model that captures quasi-steady state odor concentration profile measured without agar used for the fits in *Figure 1f and g* and (2) a

reaction-convection-diffusion model for odor-agar interaction shown in *Figure 5*. A version of this second model is also used to justify the pre-equilibration protocol, as discussed in Appendix 1.

## Convection-diffusion model for odor flow without agar

To model odor flow without agar, for example for the fits in *Figure 1f and g*, we use a 2D convection-diffusion model:

$$\frac{\partial C(x,y,t)}{\partial t} = -v\nabla C + D\nabla^2 C \tag{2}$$

where the concentration across space and time is $C(x,y,t)$, flow velocity is $v$, and the diffusion coefficient of our odor is $D$. In our chamber, at steady state ($\frac{\partial C}{\partial t} = 0$) we have:

$$v\frac{\partial C}{\partial x} = D\frac{\partial^2 C}{\partial y^2} \tag{3}$$

because with our configuration flow along the $x$-axis is dominated by convection while flow along the $y$-axis is dominated by diffusion, and therefore $\frac{\partial^2 C}{\partial x^2} \ll \frac{\partial^2 C}{\partial y^2}$.

The fit in *Figure 1f* is the solution to *Equation 3*:

$$C(x,y) = \frac{C_o}{2}(1 - (\frac{x}{2\sqrt{D\frac{x}{v}}}))\exp(-\frac{y^2}{4D\frac{x}{v}}), \tag{4}$$

where erf is the error function and $C_o$ is the odor source concentration measured in air.

In *Figure 1g* we show a fit for a 1D slice along $y$ at various positions along $x_c$, for the situation in which there is an odor source at ($y = 0, x = 0$):

$$C(y) = \frac{C(x_c, y=0)}{\sqrt{4\pi D\frac{x_c}{v}}}\exp(-\frac{y^2}{4D\frac{x_c}{v}}) \tag{5}$$

where $\frac{x_c}{v}$ is an analogy of time in non-stationary diffusion process at the cross-section at $x_c$.

For the fits in *Figure 1* the airflow velocity is set to be $v \sim 0.5$ cm/s based on the flow rate and geometry of the chamber (15 parallel tubes provide around 450 mL/min of flow into an ~255 mL chamber with ~15 cm$^2$ cross-section). The diffusion coefficient $D$ is left as a free parameter and the value that minimizes the mean-squared error between the model and the empirical measurement is used. We chose to leave the diffusion coefficient as a free parameter instead of using butanone's nominal diffusion constant of $D \sim 0.08$ cm$^2$/s, because we expect butanone's effective diffusion coefficient to be different in a confined chamber with background flow. We note that the fitted profile shown in *Figure 1g and f* and the fitted value agrees with what is expected in a stable convection-diffusion process (Peclet number $\sim 80$).

## Reaction-convection-diffusion model for odor-agar interaction

To justify the pre-equilibration protocol of *Figure 4* and to show that measurements of odor concentration along the agar's boundary allows us to infer the concentration on the agar, we propose a reaction-convection-diffusion model. This phenomenological model forms the basis of *Figure 5* and agrees with experimental observations in *Figure 6*. Compared to the convection-diffusion model, we include the 'reaction' term to account for odor-agar interactions.

The model used is a 2D generalization of this non-spatial model:

$$\frac{dC}{dt} = -\frac{1}{\tau}(C - C_o) - w\frac{dA}{dt} \tag{6}$$

$$\frac{dA}{dt} = k_a C(1 - \frac{A}{M}) - k_d A \tag{7}$$

where $C$ is a downstream concentration readout after the airflow has surface interaction with the agar gel. The influx odor concentration is $C_o$ and the odor concentration in agar is $A$. Without agar interaction, the flow chamber has its own timescale $\tau$ and the molecular flux into the agar is weighted by

a scalar $w$ (so $w = 0$ when there's no agar in the chamber). The association and dissociation constants are $k_a$ and $k_d$ and the maximum capacity of odor concentration that can be absorbed is $M$. This model is similar to the description of odorant pulse kinetics shown in *Gorur-Shandilya et al., 2019*.

In *Figure 5* we use the 2D generalization:

$$\frac{\partial}{\partial t} C(x, y) = \mathcal{L}C(x, y) - w\frac{\partial}{\partial t}A(x, y) \tag{8}$$

$$\frac{\partial}{\partial t} A(x, y) = k_a C(x, y)(1 - \frac{A(x, y)}{M(x, y)}) - k_d A(x, y) \tag{9}$$

where $\mathcal{L} = -v\nabla + D\nabla^2$ (*Equation 2*) is a linear operator for the convection-diffusion process and the odor influx is at the boundary $C(x = 0, y = 0) = C_o$. We perform numerical analysis on the set of 2D equations and permit $A$ to be non-zero only in the region where agar is present. We use a target concentration $C_o$ that is lower than $M$ and $k_a \gg k_d$ to capture odor absorption into agar. In the simulated pre-equilibration protocol we temporarily increase $C_o$ above $M$ then switch back to the target concentration to efficiently reach a steady state.

A slightly simplified version of this model forms the basis of the arguments in Appendix 1.

## Animal handling

### C. elegans

Wild-type *C. elegans* (N2) worms were maintained at 20°C on NGM agar plates with OP50 food patches. Before each chemotaxis experiments, we synchronized batches of worms and conducted measurements on young adults. Worms were rinsed with M9 solution and kept in S. Basal solution for around 30 min, while applying the pre-equilibration protocol to the flow chamber. Experiments were performed on 1.6% agar pads with chemotaxis solution (5 mM phosphate buffer with pH 6.0, 1 mM CaCl$_2$, 1 mM MgSO$_4$) (*Bargmann et al., 1993*; *Bargmann, 2006*) formed in the lid of a 96×96 mm$^2$ square dish. 50–100 worms were deposited onto the plate by pippetting down droplets of worms and removing excess solution with kimwipes. The plate was then placed in the odor flow chamber to begin recordings.

We used an integrated strain AML580 in *Figure 8*, *wtfIs491 [Pinx-1::twk-18(gf)::mcherry; Punc-122::rfp]* that expresses gain-of-function allele of *twk-18 (gf)*, an activated potassium channel, and a fluorescent protein mCherry in AIB neurons, and the fluorescent protein RFP in coelomocytes. To generate AML580, we integrated ZC2406 (yxEx1256[*Pinx-1::twk-18(gf)::mcherry; Punc-122::rfp*]) from *Luo et al., 2014*, via UV irradiation (*Evans, 2006*). These animals were out-crossed with N2 six times. AML580 has been deposited in the public Caenorhabditis Genetics Center repository at the University of Minnesota.

### D. melanogaster

Wild-type *D. melanogaster* (NM91) were maintained at 25°C incubator with 12 hr light cycle. Around 20 pairs of male and female flies were introduced into a 60 mm embryo-collection cage. A Petri dish with apple juice and yeast paste was fixed at the bottom of the cage and replaced every 3 hr for two rounds during the day time. The collected eggs were kept in the Petri dish in the same 25°C environment for another 48–60 hr to grow to second instars. We washed down and sorted out the second instar larva from the plate via 30% sucrose in water around 10 min before each behavioral experiments. We used a 96×96 mm$^2$ lid with 2.5% agar containing 0.75% activated charcoal for larval experiments (*Gepner et al., 2015*; *Gershow et al., 2012*). Around 10–20 larva were rinsed with water in a mesh and placed onto the agar plate with a paint brush. The same imaging setup and flow chamber configuration as the worm experiments were used for *Drosophila* larva.

## Imaging and behavioral analysis

### Image acquisition

Animals are imaged via a CMOS camera (Baslar, acA4112-30, with Kowa LM16FC lens) suspended above the flow chamber and illuminated by a rectangular arrangement of 850 nm LED lights. The camera acquires $2500 \times 3000$ pixel images at 14 fps. A single pixel corresponded to 32 µm on the agar plate. Labview scripts acquired images during experiments.

## *C. elegans* behavioral analysis

To increase contrast for worm imaging, a blackout fabric sheet is placed underneath the agar plate. Custom Matlab scripts based on *Liu et al., 2018*, were used to process acquired images after experiments, as linked in the Software sharing section. Briefly, the centroid position of worms was found in acquired images via thresholding and binarization. The animal's centerline was found, and its body pose was estimated following *Liu et al., 2018*, but in this work only the position and velocity was used. The tracking parameters are adjusted for this imaging setup and we extract the centroid position and velocity of worm.

The analysis pipeline focuses on the trajectory of animal navigation in the arena. The trajectories are smoothed in space with a third-order polynomial in a 0.5 s time window to remove tracking noise. We only consider tracks that appear in the recording for more than 1 min and produce displacement larger than 3 mm across the recordings. Trajectories starting at a location with odor concentration higher than 70% of the maximum odor concentration in space is removed, since these are likely tracks from animals that have performed chemotaxis already. We calculate the displacement of the center of the worm body in the camera space. The location in pixel space is aligned with the odor landscape constructed with the OSA to compute concentration gradient given a position. To avoid double counting turns when the animal turns slowly, and to mitigate effects of small displacements from tracking noise, we measure the angle change between displacement vectors over 1 s time window and define turns as angle changes larger than 60°. To quantify the curvature of navigation trajectories, we measure the angle between displacement vectors over 1 mm displacement in space.

## *D. melanogaster* behavioral analysis

Analysis of fly larvae is performed as previously (*Gepner et al., 2015*; *Gershow et al., 2012*).

## Software sharing

Software used to run the hardware and to analyze recordings are publicly available:

- https://github.com/GershowLab/OdorSensorArray copy archived at *GershowLab, 2023a*: Acquisition software that interfaces with the odor sensors; and demonstration code that regenerates many of the figures based on the data in 10.6084/m9.figshare.21737303.
- https://github.com/Kevin-Sean-Chen/leifer-Behavior-Triggered-Averaging-Tracker-new copy archived at *Kevin-SC, 2023*: Acquisition software for *C. elegans* imaging and associated analysis.
- https://github.com/GershowLab/Chen-Wu-eLife-Drosophila-Image-Capture copy archived at *GershowLab, 2023b*: Acquisition software for *Drosophila* larvae.

## Acknowledgements

Research reported in this work was supported by the National Institutes of Health National Institute of Neurological Disorders and Stroke under New Innovator award number DP2-NS116768 to AML and DP2-EB022359 to MHG; the Simons Foundation under award SCGB #543003 to AML; by the National Science Foundation, through NSF 1455015 to MHG, an NSF CAREER Award to AML (IOS-1845137), under Grant No. NSF PHY-1748958 and through the Center for the Physics of Biological Function, PHY-1734030. This work was also supported in part by the Gordon and Betty Moore Foundation Grant No. 2919.02. We thank the Kavli Institute for Theoretical Physics at University of California Santa Barbara for hosting us during the completion of this work. Strains from this work are being distributed by the CGC, which is funded by the NIH Office of Research Infrastructure Programs (P40 OD010440). We thank the Murthy Lab for flies. We thank William Jones for helpful discussion.

## Additional information

### Funding

| Funder | Grant reference number | Author |
|---|---|---|
| National Institute of Neurological Disorders and Stroke | DP2-NS116768 | Andrew M Leifer |
| National Institute of Neurological Disorders and Stroke | DP2-EB022359 | Marc H Gershow |
| Simons Foundation | SCGB #543003 | Andrew M Leifer |
| National Science Foundation | PHY-1455015 | Marc H Gershow |
| National Science Foundation | IOS-1845137 | Andrew M Leifer |
| National Science Foundation | PHY-1748958 | Marc H Gershow |
| National Science Foundation | PHY-1734030 | Andrew M Leifer |
| Gordon and Betty Moore Foundation | 2919.02 | Marc H Gershow |

The funders had no role in study design, data collection and interpretation, or the decision to submit the work for publication.

### Author contributions

Kevin S Chen, Software, Formal analysis, Investigation, Visualization, Methodology, Writing – original draft; Rui Wu, Resources, Software, Validation, Methodology, Writing – review and editing; Marc H Gershow, Conceptualization, Resources, Formal analysis, Supervision, Funding acquisition, Visualization, Methodology, Project administration, Writing – review and editing; Andrew M Leifer, Conceptualization, Supervision, Funding acquisition, Writing – original draft, Project administration, Writing – review and editing

### Author ORCIDs

Kevin S Chen http://orcid.org/0000-0001-8792-4625
Rui Wu http://orcid.org/0009-0008-8707-4075
Marc H Gershow http://orcid.org/0000-0001-7528-6101
Andrew M Leifer http://orcid.org/0000-0002-5362-5093

### Decision letter and Author response

Decision letter https://doi.org/10.7554/eLife.85910.sa1
Author response https://doi.org/10.7554/eLife.85910.sa2

## Additional files

### Supplementary files

• Supplementary file 1. Tutorial.pdf provides a step-by-step tutorial for assembling the instrument and running a typical experiment. Also available at https://doi.org/10.6084/m9.figshare.21737303.

• Supplementary file 2. PartsList_full.xlsx provides a detailed parts list. Also available at https://doi.org/10.6084/m9.figshare.21737303

• MDAR checklist

### Data availability

Recordings for odor flow control, concentration measurements, and behavioral tracking data are publicly available: https://doi.org/10.6084/m9.figshare.21737303. Compressed 'zip' archive Chen_

flow_20222.zip contains raw data. Compressed 'zip' archive flow_chamber_machined_components. zip contains CAD and other design files related to the hardware.

The following dataset was generated:

| Author(s) | Year | Dataset title | Dataset URL | Database and Identifier |
|---|---|---|---|---|
| Chen K, Gershow M, Leifer A | 2023 | Recordings for odor flow control, concentration measurements, and behavioral tracking data | https://doi.org/10.6084/m9.figshare.21737303 | figshare, 10.6084/m9.figshare.21737303 |

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

## Appendix 1

### Analytical argument for equivalence of airborne odor landscape with and without agar given equal measurements at the inlet and outlet

Without agar, the steady-state odor concentration $C_0$ obeys the following dynamics:

$$\frac{\partial}{\partial t} C_0(x, y) = \mathcal{L} C_0(x, y), \tag{10}$$

and boundary conditions,

$$\frac{\partial}{\partial y} C_0(x, y_{min}) = 0 \tag{11}$$

$$\frac{\partial}{\partial y} C_0(x, y_{max}) = 0 \tag{12}$$

$$C_0(0, y) = C_{inlet}(y) \tag{13}$$

$$C_0(L, y) = C_{outlet}(y), \tag{14}$$

where $\mathcal{L} = -v\nabla + D\nabla^2$ (**Equation 2**) is a linear operator encompassing the convection-diffusion equation. The coordinates $x, y$ are same as **Figure 4**, where the flow is along the $x$-axis and has length $L$.

With agar, assuming the same concentrations on the inlet and the outlet, nearly identical equations hold:

$$\frac{\partial}{\partial t} C_1(x, y) = \mathcal{L} C_1(x, y) + k_1 C_{agar}(x, y) - k_2 C_1(x, y) \tag{15}$$

$$\frac{\partial}{\partial t} C_{agar}(x, y) = k_2 C_1(x, y) - k_1 C_{agar}(x, y) \tag{16}$$

$$\frac{\partial}{\partial y} C_1(x, y_{min}) = 0 \tag{17}$$

$$\frac{\partial}{\partial y} C_1(x, y_{max}) = 0 \tag{18}$$

$$C_1(0, y) = C_{inlet}(y) \tag{19}$$

$$C_1(L, y) = C_{outlet}(y) \tag{20}$$

where $C_1$ is the concentration in the air and $C_{agar}$ the concentration in the agar. The first-order interactions are described with rate constants $k_1$ and $k_2$. If the air and agar are in equilibrium at all points ($k_1 C_{agar}(x, y) = k_2 C_1(x, y)$) then $C_1$ and $C_0$ obey the same equation with the same boundary conditions and are identical.

Assume that $C_1$ and $C_0$ differ by $\Delta C$:

$$C_1(x, y) = C_0(x, y) + \Delta C(x, y) \tag{21}$$

$$C_{agar}(x, y) = \frac{k_2}{k_1} C_0(x, y) + \epsilon(x, y) \tag{22}$$

then

$$\frac{\partial}{\partial t} \Delta C(x, y) = \mathcal{L} \Delta C(x, y) + k_1 \epsilon(x, y) - k_2 \Delta C(x, y) \tag{23}$$

$$\frac{\partial}{\partial t} \epsilon(x, y) = k_2 \Delta C(x, y) - k_1 \epsilon(x, y) \tag{24}$$

$$\frac{\partial}{\partial y}\Delta C(x, y_{min}) = 0 \tag{25}$$

$$\frac{\partial}{\partial y}\Delta C(x, y_{max}) = 0 \tag{26}$$

$$\Delta C(0, y) = 0 \tag{27}$$

$$\Delta C(L, y) = 0 \tag{28}$$

For $\Delta C$ to be non-zero anywhere requires that sources/sinks of excess concentration in the agar $\epsilon(x, y)$ be exactly arranged to exactly cancel out so that $\Delta C$ remains 0 on the boundary, which is unlikely. Further, the mechanism by which a non-zero $\Delta C$ would be created is by transfer of odor from regions of the agar with excess concentration ($\epsilon(x, y) > 0$ to regions with a deficit $\epsilon(x, y) < 0$), so at all points $\frac{\partial}{\partial t}|\epsilon(x, y)| < 0$ and eventually $\epsilon(x, y)$ and hence $\Delta C(x, y)$ must tend to 0 everywhere.

According to first-order kinetics, we expect the concentration change $\Delta C(x, y)$ to be proportional to the concentration at the point in space $C(x, y)$. Therefore, in practice we monitor the fractional difference as shown in *Figure 6*. When the concentration is within 10% fractional difference from the steady-state profile without agar, we declare it to be in the quasi-equilibrium regime and we proceed to conduct animal experiments.

