## [Editor Report]

In this manuscript, the authors present a valuable tool and resource to create stable odorant landscapes (gradients) for the study of chemotaxis in small model organisms. Using a metal oxide-sensor-based approach, odor concentration is measured in space and time to produce data-driven simulations of the odor diffusion process. Combined with a system of distributed odorous air flows, odorant landscapes are generated to characterize aspects of olfactory navigation in *C. elegans* and the *Drosophila* larva.

---

## [Decision Letter]

**Decision letter after peer review:**

Thank you for submitting your article "Continuous odor profile monitoring to study olfactory navigation in small animals" for consideration by *eLife*. Your article has been reviewed by 3 peer reviewers, one of whom is a member of our Board of Reviewing Editors, and the evaluation has been overseen by Piali Sengupta as the Senior Editor. The reviewers have opted to remain anonymous.

Essential revisions:

As described in their reports, the reviewers appreciate the merits of the sophisticated methodology and assays introduced in your manuscript to create controlled odorant landscapes. The reviewers raised a series of concerns related to the methodology and its use. We ask that you address these concerns in a revised manuscript.

Methodological concerns:

– The authors should experimentally characterize the sensitivity range (log concentrations over which detection is reliable), temporal resolution, and stability (potential existence of drifts) of their sensor array in response to more rapid changes in odor concentration. They should discuss their results in the context of published studies where metal oxide sensors have previously been employed to measure odor dynamics in turbulent airborne environments.

– It would be important to rule out the heat effects of the metal oxide sensor arrays on the behavior of *C. elegans* and *Drosophila* larvae. The authors should discuss the existence of such heat effects. They should also discuss whether physical interactions with the sensors themselves could affect the odor stimulus.

– The authors must generalize their conclusions to other odors that are commonly used to study olfactory behavior in *Drosophila* and *C. elegans*, and that have lower water solubility. More generally, they should discuss the type of odors that are compatible with their methodology.

– The authors should include a more thoughtful comparison of the pros and cons of their methodology with respect to the spectroscopic approach presented in Louis et al. 2008. In Tadres et al. 2022, numerical simulations of odor gradients were constrained by spectroscopy data. These simulations showed that gradients produced by a single source of ethyl butyrate are relatively stable, which is consistent with the behavior of larvae. While the reviewers don't dispute the fact that the more complex assay introduced by the authors might enhance the stability of odor gradients, it is probably misleading to conclude that all gradients produced by a single odor source are unstable.

– As supporting material, it would be important to list cost estimates to build the assay and provide general information related to the sensitivity, odor selectivity, and temporal resolution of the methodology. Given that the methodology is considered a Tools and Resources, it would also be extremely valuable to provide more information (possibly a tutorial) to help other labs adopt the methodology to characterize odor landscapes in new assays.

Additional request:

The methodology and experiments presented in the manuscript offer the opportunity to compare odor-evoked behaviors in two species in response to the same environment. To illustrate the potential of the methodology, the authors are strongly encouraged to analyze the behavior of both species using the same procedure and metrics to report similarities and differences in behavioral algorithms. This addition should help the reader define whether the methodology enables to draw new conclusions or if it recapitulates previous observations. If the authors decide that they won't proceed with a deeper behavioral analysis, they should tone down their claims about algorithmic findings in their abstract and the rest of the manuscript.

*Reviewer #1 (Recommendations for the authors):*

Measuring airborne odorant gradient is notoriously challenging. The goals of this manuscript are twofold: (1) to develop a technique to characterize the spatial properties and temporal stability of airborne odorant landscapes; (2) to produce stationary odorant landscapes with controlled and predictable geometries that are suitable to refine the study of navigation behavior (chemotaxis) in small animals such as *C. elegans* and the *Drosophila* larva. Tackling these two problems should improve future correlations between sensation and behavior to study the neural bases of sensory encoding and action selection.

To achieve their first goal, the authors propose to use a distributed array of digital gas sensors. The resolution of the measurements reported in the manuscript is impressive: they go beyond the state of the art. Although these measurements remain discrete in space, they provide valuable data to fit a biophysical model of the diffusion of the odor and its reaction with the arena where the odorant landscape is created. While the technique allows monitoring odorant landscapes in space and time, it also has limitations.

First, the measurements require modifying the experimental conditions where the behavior is tested. The landscape cannot be directly measured while the behavior is taking place. Second, the dimensions of the digital gas sensors (~1 cm) are relatively large compared to the size of the tested animals (~1 mm). This limits the resolution of the spatial measurements of odorant landscapes with small dimensions where even smaller animals are studied. In particular, the system might not be able to detect differences in concentrations between the individual inlets of the odor delivery system, which reduces the accuracy of the reconstruction of a landscape and the evaluation of its variability.

The second goal of the study is to produce odorant landscapes with enhanced stability compared to existing systems. The airflow-based olfactometer features spatially patterned odorized air. While this system represents an improvement with respect to published assays in *C. elegans*, its contribution to the larva is less obvious. The geometry of the landscapes resulting from the linear array of odorized air flows is essentially uni-dimensional. This produces geometries that are fundamentally different from the radially symmetric gradients emitted by odor cups that have been used by other labs for many years. The authors conclude that gradients resulting from the diffusion of a single source are unstable due to the absorption of the odor by the agar. This conclusion was drawn for odorant molecules that are water-soluble (or miscible in water like ethanol). Whether the same conclusion holds for odorant molecules with low or negligible water solubility is unknown, which precludes a generalization of the results to odors that have been more commonly used in the field.

The manuscript would benefit from additional work to demonstrate how the tool can advance the understanding of chemotaxis in *C. elegans* and the *Drosophila* larva. Besides the fact that the authors chose to characterize larval chemotaxis with an odor that has not been used in previous work, the behavioral characterization is limited given the potential capabilities of the new tool. Presently, it falls short of reporting "chemotaxis strategies for *C. elegans* and *D. melanogaster* larvae populations under different spatial odor landscapes" as stated in the abstract. This incompleteness is a missed opportunity to validate the merit of the methodology.

I praise the authors for the technical achievement presented in the manuscript. Combining a spatial array of digital gas sensors with computational simulations of the odor diffusion process is ingenious. However, several aspects of the methodology require further characterization and more careful validation to be helpful as a tool that can be widely adopted by other labs. Below is a list of shortcomings that should be addressed:

1. Although the multi-odorous-air-flow delivery system is simplified compared to the original design published by Gershow et al. in 2012, it remains complex. It would be important to test the reproducibility of landscapes produced by this system. In addition, it would be useful to demonstrate the flexibility with which landscapes with different geometries can be created. One notable limitation of the system is that gradients can only be designed in one direction. This precludes the creation of radially symmetric gradients produced by a point source, such as those that have been typically used in *C. elegans* (e.g. Pierce-Shimomura & Lockery 1999) or in the *Drosophila* larva (learning assay developed by Gerber and coworkers). Given the tiled arrangement of the airflow inlets, is the gradient smooth at the scale of a worm or "bumps" expected in the odor profile over the y-axis? This question is likely to be unresolved by the odor measurements since there is a higher density of tubes compared to sensors along the y-axis.

2. The manuscript reports that the high absorption of butanone by the agar significantly affects the properties of the resulting odorant landscape. As a result, odorant landscapes produced by a single odor droplet can flatten over time in such a way that the gradient might be nearly inexistent after a couple of minutes. This conclusion is significant since larval olfaction has been frequently studied with a single odor source (a cup or a droplet). As indicated by the authors in lines 270-271, butanone is a water-soluble odorant molecule. By comparison, most odors that have been used to study olfactory behaviors in the *Drosophila* larva have a low solubility in water. For instance, 1-octanol (<1g/L), benzaldehyde (<10g/L), pentyl acetate (<2g/L) and geranyl acetate (<1 g/L) are all far less soluble than butanone (275 g/L). This difference in water solubility is likely to affect the stability of the odorant gradient. Were odor measurements not combined with computational modeling of the odor diffusion-reaction process in Tadres et al. 2022 (Science Advances) to establish that gradients of ethyl butyrate and ethyl acetate are relatively stable for 5 min?

3. Technical question related to Figure 3: in this figure, the butanone droplet appears to be placed on the lid without agar (line 24) whereas it is placed directly on the agar gel in the condition with agar (line 251). The placement of the butanone droplet on the lid versus on the agar is an important distinction besides just the presence or absence of agar. This point is not addressed in the discussion of Figure 3. The authors should use the same type of odor source to make comparisons across conditions.

4. The behavioral quantification of the chemotactic behavior of *C. elegans* reproduces a series of results that have already been published. Weathervaning appears to be reproduced, but its strength might be lower than that reported by IIno and Yoshida 2009. It would be helpful if the authors could present a graph with the average curvature rate as a function of the bearing of the gradient to permit a comparison with Iino and Yoshida 2009.

In contrast with the behavioral quantification of *C. elegans*, the behavioral quantification of larvae is minimal. Essentially, the authors report that larvae turn less while moving upgradient and turn more while moving downgradient. This quantification does not justify the highly-quantitative assay introduced in the manuscript. In addition, could the authors show that larvae are capable of "weathervaning"? Another interesting application of the assay would be to establish how the navigational performances of larvae change with the steepness (or other geometric features) of stationary odorant gradients. This would also show that larvae are capable of precise chemotaxis in gradients that are essentially 1D compared to 2D gradients resulting from point sources.

5. The introduction is slightly misleading. One motivation of the work is to design a setup where measurements of odor concentrations characterizing an odorant landscape can be done during behavioral experiments. One would expect that these measurements would be done in 2D across the assay. Instead, the measurements are made on the border of the arena, which partly contradicts the original goal of the methodology. I appreciate that the presence of the sensors in the arena is not compatible with behavioral experiments, but the implications of this limitation should be more thoroughly discussed in the manuscript. It also reinforces the need to establish the level of fluctuations in odor concentrations inside the arena in real-life conditions.

More generally, how do the authors envision the use of their new methodology by regular labs working on olfaction? If, on the one hand, their primary goal is to propose a technique that will be adopted by many experimental labs, one should acknowledge that most labs might find it challenging to conduct odor measurements, optimize the convection-reaction-diffusion model and simulate new odorant landscapes. Going through the pipeline would require a "tutorial". If, on the other hand, the goal of the methodology is to draw attention on potential artifacts associated with the instability of odorant gradients in published assays, the authors should conduct a more thorough analysis of the stability of representative experimental conditions.

6. Appendix 1 argues that the convection-diffusion model and the reaction-convection-diffusion model are equivalent if the odor flow between the air and agar are in equilibrium. How is this equilibrium defined quantitatively?

*Reviewer #2 (Recommendations for the authors):*

This manuscript by Chen et al. describes an apparatus for measuring odor-evoked navigation behavior in *C. elegans* and *Drosophila* larvae. The major advance is using an array of metal oxide sensors to measure odor gradients. The authors apply this tool to generate stable gradients in an agar environment. They then measure odor-evoked behavior in both worms and fly larvae, demonstrating the ability to recover previously described stimulus-behavior associations such as biased random walking and weathervaning.

The problem of controlling and measuring odor dynamics is a challenge for all studies of odor-guided navigation and new approaches to these problems are welcome. While the present study shows the potential of the metal oxide sensor array approach, several considerations are missing that would help other researchers to evaluate whether this approach would be useful, and are important for evaluating claims made with this device.

1) Metal oxide sensors have previously been employed to measure odor dynamics in turbulent airborne environments (e.g. Schmuker et al. 2016, Tariq et al. 2021, Dinnler et al. 2022). Two major issues raised by these previous studies are that (1) metal oxide sensors have a fast onset but very slow offset which complicates inference of odor dynamics, and (2) sensor drift can complicate absolute concentration measurements. The authors should discuss these previous studies and show data on the temporal resolution and stability of their sensor array in response to more rapid changes in odor concentration.

2) Previous studies have used a spectroscopic approach to quantify odor gradients in agar (Louis et al. 2008). The authors state that this approach is not compatible with simultaneous behavioral measurements, while the metal oxide sensors are, however, it is not totally clear to me why this should be. While a direct comparison of these two methods would be ideal, a more thoughtful comparison of the pros and cons of the two methods would be most helpful to other researchers.

3) A major issue for using the metal oxide sensor arrays during behavior would seem to be the heat generated by the arrays. This should be discussed and any heat effects on behavior should be described, as thermotaxis behavior has been described in both species studied here.

4) It is not clear that this device allows for "precise" measurements at the location of the animal as claimed in the abstract. First, the measurements are in the air and are predicted to be related to the concentration in agar through a scalar. Second, measurements are made at the edges of the agar plate and internal concentrations are inferred. Although these appear to be stable in the absence of animals, local fluctuations due to animal movement cannot be measured.

Suggestions:

The focus of this study is on animals that move in a viscose substrate such as worms and larvae, where odors form stable gradients. While this is implied by the term "small model organisms" the authors should distinguish this from animals other than small model animals (adult flies, larval fish) that move in turbulent environments where the temporal resolution of the system would need to be much higher.

The ability to measure odor-evoked behavior in two species in response to the same environment seems like a bit of a missed opportunity here. Can the behavior of both species be analyzed using the same methods and similarities and differences in behavioral algorithms described?

Overall I think this is an interesting approach but I think there are a large number of specific claims that need to be softened or toned down:

line 18: "Crucially and unlike previous methods, our method allows continuous monitoring of the odor profile during behavior" I don't think this is true. For example, the Vickers and Baker 1994 study used an extra moth antenna to measure odor plume fluctuations in flying moths, and the Tariq 2021 study measured odor at the location of a navigating mouse using metal oxide sensors.

line 23-24: "accurately inferred" "precise odor concentration" not sure these are true for the reasons listed in 4 above.

line 35-36: "small mode organisms" I guess this means worms and larvae but I think it would be helpful to specifically say animals that move in a substrate or mostly navigate in gradient (as opposed to turbulent) environments.

line 46: "no technique currently exists for precise control and continuous monitoring of an odor landscape." This is not true either. Many published approaches here include optogenetics to create virtual environments, precise generation of odor waveforms, controlled flow chambers, etc.

line 97: "odor profile in the chamber" I think the abstract implies that you can make precise measurements at the location of the animal but it is clearer later on that these are inferred measurements across the arena.

line 160: "quantifying airborne odor concentrations" There is a large literature on this in the turbulent navigation field that is not cited or discussed in this paper.

line 175: "metal oxide sensors" can you give estimates of cost, sensitivity, odor selectivity, and temporal resolution? I would expect this to be in a supplement.

line 204: 1-second temporal resolution. What is the evidence for this? Is this a sampling rate or does it take into account the dynamics of the sensor as shown in the Tariq paper?

line 251: "remove two sensor bars and replace them with agar" I think it needs to be clearer upfront that the sensors have to be removed to do agar measurements.

line 371: butanone chemotaxis in *Drosophila* larvae. should cite the Jung and Bhandawat 2015 paper here that closely examines butanone-evoked navigation in adult flies and compares it to vinegar-evoked navigation.

line 399: "This last feature…sets this method apart from previous approaches" It is not clear to me why the spectroscopic approach could not in principle be used during behavior.

line 405: "In the future, such tuning curves may form the basis of investigations into neural mechanisms driving the sensorimotor transformations underlying navigation" Lots of this has been done! Seems weird to say this as a future thing and not cite the many many circuit cracking papers in worms and flies that have been pursued with other apparatus.

lines 438-440: "at the quasi-equilibrium conditions used in our experiments the odor concentration in agar is related to the airborne odor concentration directly above it up to a scalar that we predict to be constant across the agar." But is this actually true? This is quite far from the claim in the abstract that you can precisely measure the concentration at the location of the animal.

*Reviewer #3 (Recommendations for the authors):*

In this paper, Chen et al. propose a new method to measure odor stimuli in space and time. Measuring the odor stimulus is a key step in interpreting odor-driven behavior and understanding the neural mechanisms that mediate it, but this task still challenges every experimenter in the field. As described in the introduction, there is basically not a single method that is good for most behavioral assays, even when these involve small animals such as *C. elegans* and *D. melanogaster*. Previous approaches are either invasive, very expensive, or limited to very small behavioral arenas. The strength of the proposed method is to be cheap and to have a reasonable spatial (1 cm) and temporal (1 s) resolution. The full sensor array introduced here cannot be located throughout the behavioral arena, but a mathematical model shows that it is sufficient to measure the odor stimulus at certain specific positions to reconstruct the full spatial profile. In other assays, such mathematical considerations might not be possible and physical constraints might make it impossible to use the sensor during behavior, however, it can still be used to measure the odor landscape in a separate experiment, and, with a certain degree of reproducibility (which will depend on the specific delivery system), this is still better than no measure. It should be noted that the method is limited to 2D measurements, which is sufficient for the walking or crawling behavior of small animals in non-turbulent conditions, but it cannot be extended to 3D assays. It remains moreover unclear what is the sensitivity range of the sensor for the odors used in the paper (butanone and ethanol) and what is expected for other odorants (which compounds are detectable?).

As proof of principle, the authors use this new method to characterize the behavior of *C. elegans* and *D. melanogaster* as a function of the concentration gradient encountered along their moving trajectories. In this respect, it remains unclear whether the method allows new conclusions or simply recapitulates previous observations.

Sensors:

It would be important to know more about the sensitivity range of the sensors. It seems that the concentration used here does not saturate the sensors: what is the full scale of sensitivity for the x-axis in Figure 1- S1? Is the calibration curve similar for ethanol? And what kinds of odors are expected to be detectable?

Another point that is not mentioned is whether the sensor itself affects the odor stimulus: for example, through an absorption/release mechanism similar to what happens with the agar: I guess one should compare Figure 2a to Figure 4b? No interference with the stimulus would be a clear advantage of this approach over the PID and should be stated.

Behavioral analysis:

Regarding the paragraphs on *C. elegans* and *D. melanogaster*, I would suggest that the authors clarify what is a new finding vs what is already known, and in which cases the measurement of the odor gradient is critical.

Clearly, Figure 7 requires such measurements; however, the significance of the result is somehow obscure: what do we expect for the relationship between drift velocity and gradient? Also, there seems to be a very small and possibly not significant (I do not find a statistical test) positive drift for the high range of the tested gradients: could these gradients be too shallow?

Is the same analysis of drift vs gradient not possible with *D. melanogaster*? Moreover, I wonder why the turn rate for -90 and 90 degrees do not have similar values: both directions are perpendicular to the gradient, wouldn't one expect the same behavior? The heading change, in that concern, is as expected, with similar absolute value and opposite direction. Also unclear why the heading change is similar for 180 and 0 degrees.

---

## [Author Response]

Essential revisions:As described in their reports, the reviewers appreciate the merits of the sophisticated methodology and assays introduced in your manuscript to create controlled odorant landscapes. The reviewers raised a series of concerns related to the methodology and its use. We ask that you address these concerns in a revised manuscript.

We thank the reviewers and editors for this thoughtful, detailed and constructive review.

Methodological concerns:– The authors should experimentally characterize the sensitivity range (log concentrations over which detection is reliable), temporal resolution, and stability (potential existence of drifts) of their sensor array in response to more rapid changes in odor concentration. They should discuss their results in the context of published studies where metal oxide sensors have previously been employed to measure odor dynamics in turbulent airborne environments.

We performed new experiments (Figure 1 – Figure Supp 2; Figure 1 – Figure Supp 3) to characterize sensitivity range, temporal resolution and stability in response to changing odor concentrations. This is in addition to the previous experiments presented in (Figure 1 – Figure Supp 1).

Sensitivity range: Previously we had made paired recordings of the metal oxide sensor (OS) and the photoionization detector (PID) over the range of odors used in the manuscript, ~0-200 ppm (Figure 1 – Figure supp 1). Now we have added new experiments with a higher range PID (rated to 2,000 ppm) to evaluate OS responses to odors up to 1,000 ppm (Figure 1 – Figure supp 3), roughly an order of magnitude beyond the range of odors we use with animals. The OS shows non-saturating one-to-one mapping to the PID across this range.

Temporal resolution: To evaluate temporal resolution of our OS we conducted measurements of the OS in response to steps of odor concentration (Figure 1 – Figure Supp 2). We observe that both the OS and PID take tens of seconds to respond to steps in odor concentration induced by our changes to odor flow. Since our odor landscapes have nominally constant odor flow over the 30 minute timescale, we are confident that this is sufficient to capture the slow odor concentration changes that we might expect to observe.

Stability The relationship between OS and PID is stable over the course (>1 h) of our new odor step response measurements (Figure 1 – Figure supp 2), and consistent with our previous triangle wave measurements (Figure 1 – Figure Supp 1).

Thank you for pointing us to additional literature about the use of metal oxide sensors for the study of olfaction. We have now added additional references throughout:

(from Introduction):

“Metal-oxide sensors are small and scalable digital gas sensors that have recently been used to monitor odor concentration in studies of olfaction (Schmuker et al. 2016, Drix et al. 2021, Tariq et al. 2021) especially in the context of turbulent odor plumes.

Arrays of metal-oxide sensors have been used to measure spatial odor landscapes by sampling across space (Burgues et al. 2020).

Here we leverage these approaches…”

(from results):

“Metal-oxide odor sensors, designed to be used in commercial air quality sensors, are available in inexpensive and compact integrated circuit packages (Schmuker et al. 2016, Burgues et al. 2020). However, in general, commercial metal-oxide sensors are not designed for precision work – they tend to drift due to variations in heater temperature, humidity, adsorption of chemicals and ageing effects (Dennler et al. 2022). Most such sensors are designed to detect the presence of gas above a particular concentration but not to precisely measure the absolute concentration. We became aware of a newer metal-oxide sensor, the Sensirion SGP30 that was designed for long-term stability and concentration measurement; we wondered if such a sensor could be calibrated for use in an odor sensor array.”

– It would be important to rule out the heat effects of the metal oxide sensor arrays on the behavior of *C. elegans* and *Drosophila* larvae. The authors should discuss the existence of such heat effects. They should also discuss whether physical interactions with the sensors themselves could affect the odor stimulus.

We have now performed new analysis (Figure 1 – Figure supp 4) of temperature measurements to characterize heating effects of the metal oxide sensors. The existing spatial array of thermo-hydro sensors on the sensor bars already recorded the local temperature in the arena during measurements. We now present those measurements in Figure 1 – Figure Supp 4. Our measurements in an arena without agar when all 112 odor sensors are producing heat show that the temperature stabilizes after ~20 mins, a similar time frame to our pre-equilibration protocol. The spatial temperature fluctuations are small (~0.5 C) across the region where an animal would be present. In separate experiments with agar present we used a thermocouple to directly measure temperature on the agar surface and find the temperature differences to be similarly small:

“We experimentally validate heat management when the full sensor array is active for odor measurements. The temperature increases in the first 5 minutes of recording and later stabilizes (Figure 1 – Figure supp 4). After stabilizing, the temperature readout in space shows modest spatial fluctuations on the scale of 0.5 C. In separate experiments with agar, we measured similar spatial fluctuations on the agar surface.”

As we described in the caption of Figure 1 – Figure supp 4:

“We expect the heating shown here to represent an upper bound for agar experiments with animals, because here all 112 odor sensors were active and generating heat, while with agar experiments only 32 odor sensors would be active. Consistent with this, during separate experiments with agar, the temperature measured via a thermocouple at the four corners of the agar surface showed a difference in temperature within 0.4 C at four corners that are 10 cm apart.”

In the methods we describe thermal management strategies we have used to minimize the effects of heating:

“To avoid generating thermal gradients, the system has been designed to dissipate heat to the optics table. Each metal-oxide odor sensor contains a micro hotplate which consumes 86 mW power during readings. To dissipate this heat the aluminum insert inside the flow chamber serves as a heat sink. Odor sensor bars are connected to the insert using heat conductive tape and thermal paste. The insert and chamber are in turn in direct thermal contact with the optics table. Temperature and humidity is constantly monitored at 8 locations per OSB via the on-board temperature and humidity sensors during experiments to confirm that there is minimal thermal or moisture gradient created in the environment.”

Importantly, we observed little or no bias in the animal’s trajectories in control experiments that lacked odor (mock odor landscape) but still had animals, air flow and powered odor sensors, suggesting that the animals are not undergoing thermotaxis (Figure 7 – Figure Supp 1).

To test whether the odor sensor interacts with and/or depletes the odor, we conducted new experiments where we turned off the odor sensors part way through the experiment, but still recorded downstream overall odor concentration with the PID (Figure 1-Figure supp 3b). Our measurements indicate no discernible change in downstream odor concentration between when the metal-oxide odor sensors are on or off, suggesting that the metal-oxide sensors do not deplete the odor. Reviewer 3 points out that this is an expected advantage of metal-oxide sensors over PID sensors because PID sensors are expected to deplete the odor molecules that they measure.

– The authors must generalize their conclusions to other odors that are commonly used to study olfactory behavior in *Drosophila* and *C. elegans*, and that have lower water solubility. More generally, they should discuss the type of odors that are compatible with their methodology.

Thank you for the suggestion. In addition to ethanol and butanone, we now include new measurements with benzaldehyde and ethyl butyrate, two additional odor molecules that are common in larvae and worm chemotaxis studies (Figure 2 c and e). These molecules have different vapor pressure and water solubility compared to butanone (Author response table 1, Figure 2 – Figure Supp 2), but can all be measured and controlled with our apparatus and calibration protocol (new measurements in Figure 2 – Figure Supp 1).

**Author response table 1. sa2table1:** 

	Molecular weight (g/mol)	Vapor pressure (mmHg)	Water solubility (g/100 ml)
Butanone	72.11	78	27.5
Ethyl butyrate	116.16	11.3	0.586
Benzaldehyde	106.12	~1.2	0.695
Ethanol	46.06	44.6	Miscible

According to these additional measurements and the specifications provided by the manufacturer, the sensors can generally sense most volatile organic compounds of interest.

– The authors should include a more thoughtful comparison of the pros and cons of their methodology with respect to the spectroscopic approach presented in Louis et al. 2008. In Tadres et al. 2022, numerical simulations of odor gradients were constrained by spectroscopy data. These simulations showed that gradients produced by a single source of ethyl butyrate are relatively stable, which is consistent with the behavior of larvae. While the reviewers don't dispute the fact that the more complex assay introduced by the authors might enhance the stability of odor gradients, it is probably misleading to conclude that all gradients produced by a single odor source are unstable.

We have updated the text to emphasize that measurements with our sensor array do not cast doubt on the stable odor landscapes reported in Louis et al. 2008 and Tadres et al. 2022, but that our approach can also create stable landscapes with molecules that have stronger odor-agar interactions (such as butanone) or other oderants that may also interact with their environment.

“Odorants with less solubility and lower vapor pressure may not require pre-equilibration to reach quasi-equilibrium quickly, but instead may be able to stabilize across agar on a shorter time scale.

For example, ethyl butyrate is less water soluble (~0.586 g/100 ml in water, compared to 27.5 g/100 ml for butanone), has lower vapor pressure (11.3 mmHg compared to 78 mmHg for butanone) and has a smaller gas phase diffusion coefficient (5.6 x 10^-3 compared to ~69 cm^2/s for butanone). A droplet of ethyl butyrate has been shown to form a relatively stable odor landscape over a 5 minute timescale in a dish on agar without flow (Tadres et al. 2022, Louis et al. 2008). Here we demonstrate stable odor landscapes over longer timescales of at least 30 mins with a more volatile odor, butanone, by using our flow-based assay (Figure 1 – Figure Supp 5). A strength of the flow based approach with monitoring is that it should, in principle, allow for the use of a wider range of odors. Crucially, the continuous monitoring makes this approach more robust against other unforeseen odor interactions, such as interactions between an odorant and glass, aluminum or plastic.“

To facilitate direct comparison with Figure 1 in Louis et al., 2008 and Figure S10 in Tadres et al. 2022, we have added a similar analysis for odor landscape stability in Figure 1 – Figure Supp 5. We also added more explicit details about the Louis 2008 approach and how it inspires and motivates our own work.

“In one of the most comprehensive measurements to date, Louis and colleagues (Louis et al. 2008, Tadres et al. 2022) used infrared spectroscopy to measure the spatial profile of a droplet based odor gradient. They used this approach to study bilateral olfactory sensing in *Drosophila* larvae, clearly demonstrating the value of measuring odor landscapes. That work measured odor concentration at a relatively low temporal resolution (~1 sample / min), only when animals weren't present, and relied on a relatively expensive Fourier-transform infrared spectroscopy machine.”

And we draw contrasts with it and all previous works here:

“In all of these previous works, measurements were performed offline, not during animal behavior, and the odor concentration was assumed to be the same across repeats of the same experiment, and when animals are present.”

We now also discuss more explicitly a caveat with our approach: namely that it is suboptimal for simultaneous direct measurements of the full spatial gradient and animal behavior. We conducted new experiments where we demonstrate that the instrument can directly measure the odor landscape (including on the agar) simultaneously with animal behavior but that tracking then becomes a challenge (Figure 1 – Figure Supp 6). We describe this in more detail in response to Reviewer #1 and mention it in the Results section:

“Ideally, one would like to directly measure odor opposite the surface of the agar, such that odor-agar-interactions are fully accounted for. By placing the agar on the lid hanging above the odor sensors, we were able to simultaneously record spatial odor concentrations generated by our flow chamber opposite the surface of the agar during animal behavior (Figure 1 – Figure Supp 6). Unfortunately, when imaging in this configuration, the odor sensors in the background obstruct the animals from being tracked over the extent of the arena. We therefore sought an alternative strategy: we chose to measure the odor profile along only the boundary of the arena in order to infer the odor concentration inside the arena in such a way as to also take into account the odor-agar-interactions…”

And in the discussion:

“Here we also showed the possibility of and challenges associated with directly measuring the full spatial odor profile (instead of using the boundary) during simultaneous behavior tracking (Figure 1 – Figure Supp 6). Future work is needed to improve concurrent animal tracking methods or to design more miniaturized and scalable sensors.”

– As supporting material, it would be important to list cost estimates to build the assay and provide general information related to the sensitivity, odor selectivity, and temporal resolution of the methodology. Given that the methodology is considered a Tools and Resources, it would also be extremely valuable to provide more information (possibly a tutorial) to help other labs adopt the methodology to characterize odor landscapes in new assays.

We thank the reviewer for this suggestion. We now include cost breakdowns in Table 1 in the methods section. We continue to provide comprehensive parts and catalog numbers (PartsList.xlsx), now as Supplementary File 2. And we provide a new detailed step-by-step tutorial (Tutorial.pdf, Supplementary File 1) with photographs and instructions to assemble and operate the instrument. Both Supplementary Files are also available in a FigShare repository, http://doi.org/10.6084/m9.figshare.21737303.

Additional request:The methodology and experiments presented in the manuscript offer the opportunity to compare odor-evoked behaviors in two species in response to the same environment. To illustrate the potential of the methodology, the authors are strongly encouraged to analyze the behavior of both species using the same procedure and metrics to report similarities and differences in behavioral algorithms. This addition should help the reader define whether the methodology enables to draw new conclusions or if it recapitulates previous observations. If the authors decide that they won't proceed with a deeper behavioral analysis, they should tone down their claims about algorithmic findings in their abstract and the rest of the manuscript.

We have now added new experiments and a new maintext figure characterizing chemotaxis of a mutant worm with ectopic expression of an activated potassium channel (twk-18(gf)) expressed in the interneuron AIB (Figure 8). We compare their navigation to wild type and show that these mutant animals still climb gradients but they have lower turning rate and rely less on weather-vaning. This comparative analysis demonstrates how our setup can provide insight into potential neural circuit mechanisms that contribute to navigation behavior in worms. We separately have added additional language comparing worms and *Drosophila* larvae:

“We note that the chemotaxis efficiency is much higher in larvae compared to worms given the order of magnitude difference in run velocity (~ 3 mm/s in larvae and ~0.1 mm/s in worms).”

Reviewer #1 (Recommendations for the authors):Measuring airborne odorant gradient is notoriously challenging. The goals of this manuscript are twofold: (1) to develop a technique to characterize the spatial properties and temporal stability of airborne odorant landscapes; (2) to produce stationary odorant landscapes with controlled and predictable geometries that are suitable to refine the study of navigation behavior (chemotaxis) in small animals such as *C. elegans* and the Drosophila larva. Tackling these two problems should improve future correlations between sensation and behavior to study the neural bases of sensory encoding and action selection.To achieve their first goal, the authors propose to use a distributed array of digital gas sensors. The resolution of the measurements reported in the manuscript is impressive: they go beyond the state of the art. Although these measurements remain discrete in space, they provide valuable data to fit a biophysical model of the diffusion of the odor and its reaction with the arena where the odorant landscape is created. While the technique allows monitoring odorant landscapes in space and time, it also has limitations.First, the measurements require modifying the experimental conditions where the behavior is tested. The landscape cannot be directly measured while the behavior is taking place. Second, the dimensions of the digital gas sensors (~1 cm) are relatively large compared to the size of the tested animals (~1 mm). This limits the resolution of the spatial measurements of odorant landscapes with small dimensions where even smaller animals are studied. In particular, the system might not be able to detect differences in concentrations between the individual inlets of the odor delivery system, which reduces the accuracy of the reconstruction of a landscape and the evaluation of its variability.

We thank the reviewer for this thoughtful critique. We now demonstrate direct recording of the entire extent of the odor landscapes when agar is present in the flow chamber (Figure 6) and when animals are also present on agar (Figure 1 – Figure Supp 6). We show that the current system, as designed, has challenges tracking animals when the sensors are present, and we now explain how this motivated us to pursue a strategy of measuring only on the boundary when animals are present.

We analyzed the subset of animals that we are able to track simultaneously with odor measurements and show that chemotaxis strategies qualitatively agree with the findings presented in Figure 7.

Regarding the spatial resolution of our odor measurements, we are confident that our sensors are sufficient to capture spatial variation for the geometries and flows used throughout this manuscript which are all far from the turbulent regime. Inlets are spaced 1 cm apart while our sensors are spaced 0.75 cm apart vertically along the sensor bar. We now clarify in the revised methods section:

“The 16 odor sensors are arranged in two interleaved columns spaced 1 cm apart. The vertical distance between sensors in each column is 1.5 cm, which results in 0.75 cm vertical sensor resolution.”

Crucially, our model-based fitting indicates that the flow patterns we use are governed by a non-turbulent convection-diffusion process, which has a spatial scale of ~1 cm (Figure 1 g). Therefore we are confident that we have sufficient spatial resolution. We also are reassured that we see stability in the time domain. We present new analysis (Figure 1 – Figure Supp 5) to show that our sensor size and spacing is sufficient to capture the spatial concentration changes we expect in the portion of the arena that we measure over time. Our measurements show that there is little temporal fluctuation throughout the time scale of behavioral recordings (30 minutes) and that the spatial profile is smooth and robust in its shape. We therefore conclude that the current setup should be sufficient to characterize the spatially patterned odor landscape.

The second goal of the study is to produce odorant landscapes with enhanced stability compared to existing systems. The airflow-based olfactometer features spatially patterned odorized air. While this system represents an improvement with respect to published assays in *C. elegans*, its contribution to the larva is less obvious. The geometry of the landscapes resulting from the linear array of odorized air flows is essentially uni-dimensional. This produces geometries that are fundamentally different from the radially symmetric gradients emitted by odor cups that have been used by other labs for many years. The authors conclude that gradients resulting from the diffusion of a single source are unstable due to the absorption of the odor by the agar. This conclusion was drawn for odorant molecules that are water-soluble (or miscible in water like ethanol). Whether the same conclusion holds for odorant molecules with low or negligible water solubility is unknown, which precludes a generalization of the results to odors that have been more commonly used in the field.

We apologize for the confusion. We did not mean to imply that single source droplets are unstable for all odor molecules or all cases. We also address this point in more detail in response to the Essential Revisions above. We now clarify in the text:

“We focused on the odor butanone because it is important for a prominent associative learning assay (Torayama et al. 2007, Kauffman et al.2011). Butanone has high vapor pressure and is highly soluble in water and therefore it interacts strongly with agar. In this work we show that this odor-agar interaction makes it difficult to a priori infer an odor landscape experienced by the animal when agar is present, but that continuously monitoring the odor profile on the boundary overcomes this challenge.”

…

“Odorants with less solubility and lower vapor pressure may not require pre-equilibration to reach quasi-equilibrium quickly, but instead may be able to stabilize across agar on a shorter time scale.

For example, ethyl butyrate is less water soluble (~0.586 g/100 ml in water, compared to 27.5 g/100 ml for butanone), has lower pressure (11.3 mmHg compared to 78 mmHg for butanone) and has a smaller gas phase diffusion coefficient (5.6 x 10^-3 cm^2/s compared to ~69 cm^2/s for butanone). A droplet of ethyl butyrate has been shown to form a relatively stable odor landscape over a 5 minute timescale in a dish on agar without flow (Tadres et al. 2022, Louis et al. 2008).”

And we have clarified in the relevant subheading (emphasis added): “Odor-agar interactions dominate classical *butanone* droplet assays”

We now also conducted additional experiments, in new Figure 2c and e, to show that our olfactometer is capable of making gradients of three other odors with other properties: benzaldehyde, ethanol and ethyl butyrate. In principle, molecules with lower water solubility should be easier to work with because they interact less with agar. A strength of our approach is that it works with a range of odor molecules including ones like butanone that we are interested in for scientific reasons but that have high interactions with agar.

The manuscript would benefit from additional work to demonstrate how the tool can advance the understanding of chemotaxis in *C. elegans* and the *Drosophila* larva. Besides the fact that the authors chose to characterize larval chemotaxis with an odor that has not been used in previous work, the behavioral characterization is limited given the potential capabilities of the new tool. Presently, it falls short of reporting "chemotaxis strategies for *C. elegans* and *D. melanogaster* larvae populations under different spatial odor landscapes" as stated in the abstract. This incompleteness is a missed opportunity to validate the merit of the methodology.

We have now removed the offending language from the abstract. Importantly, we have added new experiments to demonstrate how the tool can advance our understanding of chemotaxis. In particular we now perform new experiments comparing a mutant worm that has defects in a key interneuron AIB to a wild-type animal (Figure 8). These measurements show that defects in AIB result in a down-regulation of weathervaning, and lower overall turning rate, but that the bias in turning rate is preserved and the animal still climbs odor gradients. Experiments like the one with the AIB mutant show that our instrument will enable new advances. We think the work is particularly well suited to its “Tools and Resources” format, as described in the *eLife* guide for authors:

“Tools and Resources articles do not have to report major new biological insights or mechanisms, but it must be clear that they will enable such advances to take place”.

I praise the authors for the technical achievement presented in the manuscript. Combining a spatial array of digital gas sensors with computational simulations of the odor diffusion process is ingenious. However, several aspects of the methodology require further characterization and more careful validation to be helpful as a tool that can be widely adopted by other labs. Below is a list of shortcomings that should be addressed:1. Although the multi-odorous-air-flow delivery system is simplified compared to the original design published by Gershow et al. in 2012, it remains complex. It would be important to test the reproducibility of landscapes produced by this system. In addition, it would be useful to demonstrate the flexibility with which landscapes with different geometries can be created. One notable limitation of the system is that gradients can only be designed in one direction. This precludes the creation of radially symmetric gradients produced by a point source, such as those that have been typically used in *C. elegans* (e.g. Pierce-Shimomura & Lockery 1999) or in the *Drosophila* larva (learning assay developed by Gerber and coworkers). Given the tiled arrangement of the airflow inlets, is the gradient smooth at the scale of a worm or "bumps" expected in the odor profile over the y-axis? This question is likely to be unresolved by the odor measurements since there is a higher density of tubes compared to sensors along the y-axis.

The reviewer raises several related concerns about (1) reproducibility (2) limitations of possible geometries and (3) spatial resolution.

1)We demonstrate several aspects of the instrument's reproducibility. Perhaps the most dramatic form of reproducibility is that we show in Figure 6 that the system reproduces a desired odor landscape using the same steady-state flow inputs even when a large agar odor sink is present that would otherwise dramatically perturb the landscape. Specifically Figure 6b generates the same odor landscape as in Figure 6a to within 10% fractional difference even though the agar pad is present. We also show Figure 1 – Figure Supp 1 that the odor sensors themselves are reproducible across time. And in a new analysis in Figure 1 – Figure Supp 5 we show that the odor gradient itself is stable over time.2)Regarding concerns about possible geometry: We argue that the droplet assay is more limited than the geometries allowed by our instrument because the droplet assay allows only a single odor landscape geometry and the experimenter has limited control over its properties. In contrast, our instrument can support and measure a droplet based gradient, as we show in Figure 3. So if a droplet is desired, our instrument can measure from that too. But importantly our instrument can also use flow to generate other odor landscapes such as a cone with a different slope or inverse cone as shown in Figure 2. As an aside, in the *C. elegans* literature and in some early fly larva experiments the radial symmetry of the droplet assay is often not used because chemotaxis arrays are conducted when the droplets are positioned on the far side of a plate (Bargmann 2006, Cho et al. 2016, Aceves-Piña et al. 1979).3)We address concerns about spatial resolution of our odor sensors in more detail in response to Reviewer #1’s Public Review, above. The inter-channel spacing (1 cm) is in fact larger than the inter-sensor spacing (0.75 cm vertical sensor resolution). Fine structure due to odor plumes is ruled out by the Reynold’s number (<10), which is too low to support turbulent flow, and by quantitative agreement of the measurements with the convective-diffusive model lacking turbulence. This model is characterized by a length scale larger than the inter-sensor spacing. Therefore we are confident that we have the spatial resolution to characterize the relevant changes of the landscape.

2. The manuscript reports that the high absorption of butanone by the agar significantly affects the properties of the resulting odorant landscape. As a result, odorant landscapes produced by a single odor droplet can flatten over time in such a way that the gradient might be nearly inexistent after a couple of minutes. This conclusion is significant since larval olfaction has been frequently studied with a single odor source (a cup or a droplet). As indicated by the authors in lines 270-271, butanone is a water-soluble odorant molecule. By comparison, most odors that have been used to study olfactory behaviors in the *Drosophila* larva have a low solubility in water. For instance, 1-octanol (<1g/L), benzaldehyde (<10g/L), pentyl acetate (<2g/L) and geranyl acetate (<1 g/L) are all far less soluble than butanone (275 g/L). This difference in water solubility is likely to affect the stability of the odorant gradient. Were odor measurements not combined with computational modeling of the odor diffusion-reaction process in Tadres et al. 2022 (Science Advances) to establish that gradients of ethyl butyrate and ethyl acetate are relatively stable for 5 min?

We agree that water solubility plays an important role. We now clarify in the main text (copied below) that our claims about the instability of butanone droplets apply to butanone and not necessarily any other odorant. We also address this concern in response to Essential Revisions (third bullet point) and in response to Reviewer #1’s Public Review.

“Odorants with less solubility and lower vapor pressure may not require pre-equilibration to reach quasi-equilibrium quickly, but instead may be able to stabilize across agar on a shorter time scale. For example, ethyl butyrate is less water soluble (~0.586 g/100 ml in water, compared to 27.5 g/100 ml for butanone), has lower pressure (11.3 mmHg compared to 78 mmHg for butanone) and has a smaller gas phase diffusion coefficient (5.6 x 10^-3 cm^2/s compared to ~69 cm^2/s for butanone). A droplet of ethyl butyrate has been shown to form a relatively stable odor landscape over a 5 minute timescale in a dish on agar without flow (Tadres et al. 2022, Louis et al. 2008). Here we demonstrate stable odor landscapes over longer timescales of at least 30 mins with a more volatile odor, butanone, by using our flow-based assay (Figure 1 – Figure Supp 5).”

We are not casting doubt on the claims made in Tadres et al. 2022. It is interesting to compare the stability of the odor landscapes in Tadres et al. to the ones shown here. For the sake of comparison we copy Figure S10 from Tadres et al. and show it next to a panel from our new Figure 1 – Figure Supp 5. Even with a much less stable odorant our flow based approach demonstrates stable odor landscapes that persist five-times longer.

This prolonged stability would be important for research in learning, sensory adaptation, or any chemotaxis experiments in worms, which take longer than for flies.

3. Technical question related to Figure 3: in this figure, the butanone droplet appears to be placed on the lid without agar (line 24) whereas it is placed directly on the agar gel in the condition with agar (line 251). The placement of the butanone droplet on the lid versus on the agar is an important distinction besides just the presence or absence of agar. This point is not addressed in the discussion of Figure 3. The authors should use the same type of odor source to make comparisons across conditions.

Thank you for the suggestion. We conducted the requested experiments. We now show both droplets on the lid and droplet on agar in revised Figure 3. It's perhaps an interesting historical tradition that odor droplets are normally placed in agar for worm chemotaxis (Bargemann 2006; Cho et al. 2016) and on the lid for larval chemotaxis experiments (Tadres et al. 2022 and Louis et al. 2008).

4. The behavioral quantification of the chemotactic behavior of *C. elegans* reproduces a series of results that have already been published. Weathervaning appears to be reproduced, but its strength might be lower than that reported by IIno and Yoshida 2009. It would be helpful if the authors could present a graph with the average curvature rate as a function of the bearing of the gradient to permit a comparison with Iino and Yoshida 2009.

Thank you for the suggestion. We generate a new Figure 7–Figure Supp 2 to more directly compare weathervaning in butanone chemotaxis to the salt chemotaxis reported by Iino and Yoshida 2009. As shown in Iino 2009 figure 4B, their results with isoamyl alcohol show average curving rate on a similar scale (they do not have direct odor concentration measurements). Our result not only demonstrates weathervaning strategy in butanone chemotaxis but also indicates that the tuning curve depends on the magnitude of the gradient (as opposed to just the bearing angle).

In contrast with the behavioral quantification of *C. elegans*, the behavioral quantification of larvae is minimal. Essentially, the authors report that larvae turn less while moving upgradient and turn more while moving downgradient. This quantification does not justify the highly-quantitative assay introduced in the manuscript. In addition, could the authors show that larvae are capable of "weathervaning"? Another interesting application of the assay would be to establish how the navigational performances of larvae change with the steepness (or other geometric features) of stationary odorant gradients. This would also show that larvae are capable of precise chemotaxis in gradients that are essentially 1D compared to 2D gradients resulting from point sources.

These are excellent suggestions for a future larger scale investigation. In this work we demonstrate that the instrument will enable future discoveries for both worms and fly larvae. For example we show the diversity of odor landscapes (e.g. Figure 2) that one could use to address the scientific questions like those posed by the reviewer. We demonstrate that larvae navigate in the new apparatus as expected and as previously reported, supporting its use in the sorts of future experiments described by the reviewer.

5. The introduction is slightly misleading. One motivation of the work is to design a setup where measurements of odor concentrations characterizing an odorant landscape can be done during behavioral experiments. One would expect that these measurements would be done in 2D across the assay. Instead, the measurements are made on the border of the arena, which partly contradicts the original goal of the methodology. I appreciate that the presence of the sensors in the arena is not compatible with behavioral experiments, but the implications of this limitation should be more thoroughly discussed in the manuscript. It also reinforces the need to establish the level of fluctuations in odor concentrations inside the arena in real-life conditions.

We now include new measurements of the full sensor array concurrently with agar gel (Figure 6) and worm behavior (Figure 1 – Figure Supp 6). The new measurements improve the manuscript in several ways: First, they show that simultaneous direct measurements are possible, although suboptimal because they create challenges for behavioral tracking shown in Figure 1 – Figure Supp 6. Second, they provide strong motivation for moving to a boundary based approach, which we now explain more clearly to the reader. And third, the agar measurements in Figure 6 provide additional empirical evidence that our model (Figure 5 and Appendix) and boundary conditions (Figure 4) are effective for inferring concentration across the 2D space during behavioral recordings. And the measurements in Figure 6 provide direct evidence for a stable, controlled odor landscape in our system.

We describe the textual changes we have made in more detail in our response to Reviewer #1’s Public Review, and in our response to the requested Essential Revisions.

More generally, how do the authors envision the use of their new methodology by regular labs working on olfaction? If, on the one hand, their primary goal is to propose a technique that will be adopted by many experimental labs, one should acknowledge that most labs might find it challenging to conduct odor measurements, optimize the convection-reaction-diffusion model and simulate new odorant landscapes. Going through the pipeline would require a "tutorial". If, on the other hand, the goal of the methodology is to draw attention on potential artifacts associated with the instability of odorant gradients in published assays, the authors should conduct a more thorough analysis of the stability of representative experimental conditions.

Our goal is to provide a resource to the field. Our goal is not to systematically explore potential artifacts in other systems. We chose to include measurements of the butanone odor droplet (Figure 3) in order to motivate the need for new methods to better deal with the odor butanone, which we are interested in using for studying learned olfactory navigation.

We have made several changes throughout the text to make clear that we are not casting doubt on the previous measurements, such as those in Tadres et al. 2022 and Louis et al. 2008. We describe these textual changes in more detail in response to requested Essential Revisions, Reviewer #1’s public review, and Reviewer #1 point 2, above.

There may be a misunderstanding about the convection-reaction-diffusion model. That model is to verify our protocol to a reader, but is NOT required for the experiment. No modeling is needed for a new lab to adopt this method and build their own custom stationary odor gradients, as we now state:

“Note our hypothetical experimenter would not be required to perform any simulations or calculations.”

A lab needs only to construct stationary landscapes without agar, and then ensure that the odor measured along the boundary is similar when an agar plate is introduced. We now state this explicitly in the main text to avoid confusion:

“Note that the experimental procedure does not require one to conduct numerical simulations repetitively. According to the empirical (Figure 6) and theoretical (Appendix) evidence, one can confidently infer the odor landscape from boundary conditions when the system reaches quasi-equilibrium.”

…

“A detailed tutorial for assembling the instrument and the protocol for experiments is provided in Supplementary File 1.”

We also now include a step-by-step tutorial, with photographs, to help other labs perform experiments with the instrument. The tutorial is included as Supplementary File 1, and crossposted on FigShare https://doi.org/10.6084/m9.figshare.21737303.

6. Appendix 1 argues that the convection-diffusion model and the reaction-convection-diffusion model are equivalent if the odor flow between the air and agar are in equilibrium. How is this equilibrium defined quantitatively?

We now clarify in the appendix:

“According to first-order kinetics, we expect the concentration change \Δ C(x,y) to be proportional to the concentration at the point in space C(x,y). Therefore in practice we monitor the fractional difference as shown in (Figure 6). Quantitatively when the concentration is within 10% fractional difference from the steady-state profile without agar, we categorize it as the quasi-equilibrium regime we conduct animal experiments with.”

We define equilibrium between air and agar when the concentration difference in two substrates has little change in time, namely k1\deltaC ~ k2\epsilon. Empirically and quantitatively, when the outflux odor concentration under conditions with or without agar is close (within a given fraction that the experimentalist chooses), then we conduct animal experiments in this quasi-steady-state regime. In our revised manuscript, we have new measurements for odor concentration across the agar region (Figure 6). The fractional difference is below 10% at equilibrium in our system.

Reviewer #2 (Recommendations for the authors):This manuscript by Chen et al. describes an apparatus for measuring odor-evoked navigation behavior in *C. elegans* and *Drosophila* larvae. The major advance is using an array of metal oxide sensors to measure odor gradients. The authors apply this tool to generate stable gradients in an agar environment. They then measure odor-evoked behavior in both worms and fly larvae, demonstrating the ability to recover previously described stimulus-behavior associations such as biased random walking and weathervaning.The problem of controlling and measuring odor dynamics is a challenge for all studies of odor-guided navigation and new approaches to these problems are welcome. While the present study shows the potential of the metal oxide sensor array approach, several considerations are missing that would help other researchers to evaluate whether this approach would be useful, and are important for evaluating claims made with this device.1) Metal oxide sensors have previously been employed to measure odor dynamics in turbulent airborne environments (e.g. Schmuker et al. 2016, Tariq et al. 2021, Dinnler et al. 2022). Two major issues raised by these previous studies are that (1) metal oxide sensors have a fast onset but very slow offset which complicates inference of odor dynamics, and (2) sensor drift can complicate absolute concentration measurements. The authors should discuss these previous studies and show data on the temporal resolution and stability of their sensor array in response to more rapid changes in odor concentration.

We thank the reviewer for pointing us to this literature, which we now cite at several points in the manuscript, as described in response to the first requested Essential Revision. To characterize temporal dynamics, we now present new measurements in response to step-wise odor concentration changes (Figure 1 – Figure Supp 2). We find the response timescale of 10’s-of-seconds to be sufficient for our odor landscapes which are nominally static over the timescale of our recordings (30 mins). We also characterize temporal drift across 90 minutes in Figure 1 – Figure Supp 1 by comparing the metal oxide sensors to a PID. We see no obvious systematic drift on this timescale. Some of those earlier metal-oxide studies involve detecting odor in the turbulent regime which is much more demanding than the odor landscapes used here

We now highlight these measurements and comparisons to previous uses of metal oxide sensors in the text:

“Metal-oxide sensors are small and scalable digital gas sensors that have recently been used to monitor odor concentration in studies of olfaction (Schmuker et al. 2016, Drix et al. 2021, Tariq et al. 2021) especially in the context of turbulent odor plumes. Arrays of metal-oxide sensors have been used to measure spatial odor landscapes by sampling across space (Burgues et al. 2020). Here we leverage these approaches to present a new multi-sensor odor array with metal-oxide sensors combined with a flow chamber”.

…

“However, in general, commercial metal-oxide sensors are not designed for precision work – they tend to drift due to variations in heater temperature, humidity, adsorption of chemicals and ageing effects (Dennler et al. 2022).”

2) Previous studies have used a spectroscopic approach to quantify odor gradients in agar (Louis et al. 2008). The authors state that this approach is not compatible with simultaneous behavioral measurements, while the metal oxide sensors are, however, it is not totally clear to me why this should be. While a direct comparison of these two methods would be ideal, a more thoughtful comparison of the pros and cons of the two methods would be most helpful to other researchers.

We do not claim that the FTIR method is fundamentally incompatible with simultaneous behavioral measurements, merely that it has not been demonstrated before. We have now provided a more detailed comparison in the introduction:

“In one of the most comprehensive measurements to date, Louis and colleagues (Louis et al. 2008, Tadres et al. 2022) used infrared spectroscopy to measure the spatial profile of a droplet based odor gradient. They used this approach to study bilateral olfactory sensing and olfactory encoding in *Drosophila* larvae, clearly demonstrating the value of measuring odor landscapes. That work measured odor concentration at a relatively low temporal resolution (~1 sample / min), only when animals weren't present, and relied on a relatively expensive Fourier-transform infrared spectroscopy machine.”

…

“In all of these previous works, measurements were performed offline, not during animal behavior, and the odor concentration was assumed to be the same across repeats of the same experiment, and when animals are present.”

3) A major issue for using the metal oxide sensor arrays during behavior would seem to be the heat generated by the arrays. This should be discussed and any heat effects on behavior should be described, as thermotaxis behavior has been described in both species studied here.

We thank the reviewer for pointing out the importance of heat management. We address this in more detail in response to the second point in “Essential revisions.” Briefly, we added new analysis for heating (Figure 1 – Figure Supp 4) and more detailed description in the caption and method sections. Specifically, we leverage existing spatial temperature measurements made by thermo-hydro sensors located across the odor sensor array. The metal-oxide sensors produce heat during the experiment but we dissipate much of this heat through thermal coupling to the optics table. We also observe no obvious signs of thermotaxis or biased trajectories in control experiments where we would expect any heating to be present (Figure 7 – Figure Supp 1). We therefore conclude that the heating effects are minor.

4) It is not clear that this device allows for "precise" measurements at the location of the animal as claimed in the abstract. First, the measurements are in the air and are predicted to be related to the concentration in agar through a scalar. Second, measurements are made at the edges of the agar plate and internal concentrations are inferred. Although these appear to be stable in the absence of animals, local fluctuations due to animal movement cannot be measured.

We have removed the offending language from the text and added new experiments that directly measure odor concentration above the agar (Figure 6) even when animals are present (Figure 1 – Figure Supp 6). This direct measurement validates our approach of inferring internal concentrations from the boundary. We describe this in more detail in response to Reviewer #1’s public review, and in response to the requested Essential Revisions. We now write in the text:

“Ideally, one would like to directly measure odor just opposite the surface of the agar, such that odor-agar-interactions are fully accounted for. By placing the agar on the underside of the lid hanging above the odor sensors, we were able to simultaneously record spatial odor concentrations generated by our flow chamber opposite the surface of the agar during animal behavior (Figure 1 – Figure Supp 6). Unfortunately, when imaging in this configuration, the odor sensors in the background obstruct the animals from being tracked over the extent of the arena. We therefore sought an alternative strategy: we chose to measure the odor profile along only the boundary of the arena in order to infer the odor concentration inside the arena in such a way as to also take into account the odor-agar-interactions.”

In experiments, such as new measurements in Figure 6 – Figure Supp 1, we see no obvious evidence to suggest that the movement of animals significantly alters the odor landscape:

"After the pre-equilibrium protocol, measured airborne concentration in the arena with agar was similar to measurements made without agar to within an average of less than 10% fractional difference between the two conditions in the area that the animal would crawl, FIGURE 6b. The agreement of odor landscape measured with and without agar at quasi-equilibrium agrees with our modeling and indicates that the odor landscape is stable and identifiable in our setup by monitoring the odor profile along the boundary…

The presence of worms did not noticeably alter the odor concentration (Figure 6 – Figure Supp 1)."

Suggestions:The focus of this study is on animals that move in a viscose substrate such as worms and larvae, where odors form stable gradients. While this is implied by the term "small model organisms" the authors should distinguish this from animals other than small model animals (adult flies, larval fish) that move in turbulent environments where the temporal resolution of the system would need to be much higher.

Thank you for this suggestion. To further clarify that the method is developed for stable gradients, we now write:

“Here we have addressed the problem of creating non-turbulent airborne odor landscapes.”

In the abstract we make clear we discuss worms and fly larvae:

“For small model organisms like *C. elegans* and larval *D. melanogaster*…”

and that we deal with stationary landscapes:

“We construct stationary chemical landscapes in an odor flow chamber…”

The ability to measure odor-evoked behavior in two species in response to the same environment seems like a bit of a missed opportunity here. Can the behavior of both species be analyzed using the same methods and similarities and differences in behavioral algorithms described?

We have now expanded the language comparing the navigational strategy of these two organisms in the same environment:

“With modulation of turning rate and orientation in the same odor environment, we conclude that *Drosophila* larvae use similar navigational strategies to *C. elegans* to move towards butanone. We note that the chemotaxis efficiency is much higher in larvae compared to worms given the order of magnitude difference in run velocity (~3 mm/s in larvae and ~ 0.1 mm/s in worms).”

Overall I think this is an interesting approach but I think there are a large number of specific claims that need to be softened or toned down:line 18: "Crucially and unlike previous methods, our method allows continuous monitoring of the odor profile during behavior" I don't think this is true. For example, the Vickers and Baker 1994 study used an extra moth antenna to measure odor plume fluctuations in flying moths, and the Tariq 2021 study measured odor at the location of a navigating mouse using metal oxide sensors.

Thank you for pointing us to this literature. We removed the offending phrase from the abstract and now highlight the prior work:

“A central difficulty in measuring animals' responses to olfactory cues is quantifying airborne odor concentrations that vary in space and time (Vickers et al. 1994, Tariq et al. 2021, Kadakia et al. 2022, Yamazoe-Umemoto et al. 2018).”

line 23-24: "accurately inferred" "precise odor concentration" not sure these are true for the reasons listed in 4 above.

We now show more direct evidence to support our claims that our inference is accurate to within 10% (Figure 6), and we have also removed the offending language. We discuss this in more detail in response to Reviewer #2’s Public Review point #4, above.

line 35-36: "small mode organisms" I guess this means worms and larvae but I think it would be helpful to specifically say animals that move in a substrate or mostly navigate in gradient (as opposed to turbulent) environments.

We now clarify in the introduction:

“We focus in particular on the navigation of small model organisms in continuous, non-turbulent, gradients established by the spread of odorants from their sources due to diffusion and drift.”

line 46: "no technique currently exists for precise control and continuous monitoring of an odor landscape." This is not true either. Many published approaches here include optogenetics to create virtual environments, precise generation of odor waveforms, controlled flow chambers, etc.

We removed the original claim and revised the introduction section to include methods used in past studies, including virtual environments and controlled odor presentation:

“While many techniques exist to present odor stimuli (Gorur-Shandilya et al. 2019, Kadakia et al. 2022), measure odor concentration (Tariq et al. 2021, Celani et al. 2014), or develop virtual odor environments in a lab (Yamada et al. 2021, Radvansky et al. 2018), it remains challenging for precise control and continuous monitoring of an odor landscape.”

Optogenetic virtual environments are not a direct substitute for precise odor environments because optogenetics bypasses the biophysics that govern receptor binding and may or may not evoke sensory adaptation in a physiological way.

line 97: "odor profile in the chamber" I think the abstract implies that you can make precise measurements at the location of the animal but it is clearer later on that these are inferred measurements across the arena.

We have reworded the abstract to further clarify:

“Careful placement of the sensors allows the odor concentration across the arena to be continuously inferred in space and monitored through time.”

Moreover, we now show direct measurements of the odor profile in the chamber in new experiments in Figure 6 and Figure 1 – Figure Supp 6.

line 160: "quantifying airborne odor concentrations" There is a large literature on this in the turbulent navigation field that is not cited or discussed in this paper.

We thank the reviewer for pointing us to these references. We now discuss them:

“A central difficulty in measuring animals' responses to olfactory cues is quantifying airborne odor concentrations that vary in space and time (Vckers et al. 1994, Tariq et al. 2021, Kadakia et al. 2022, Yamazoe-Umemoto et al. 2018). This difficulty is exacerbated in turbulent environments where odor plumes carry abrupt spatial and temporal jumps in concentration far from the source with fundamentally unpredictable dynamics (Celani et al. 2014, Dennler et al. 2022)”

line 175: "metal oxide sensors" can you give estimates of cost, sensitivity, odor selectivity, and temporal resolution? I would expect this to be in a supplement.

Table 1 now lists the cost of key components. At the time of purchase, the Metal Oxide Sensors were approximately $10 each. We also provide a manifest of detailed parts numbers (Supplementary File 2, PartsList.xlsx, cross listed on FigShare *10.6084/m9.figshare.21737303*).

We address reviewer concerns about sensitivity, selectivity and temporal resolution in more detail in our response to the first requested “Essential revision”. Briefly, we conducted new measurements to characterize their responses and stability presented in new figure panels (Figure 1 – Figure Supp 2; Figure 1 – Figure Supp 3; and Figure 2):

Sensitivity: We added calibration with odor sensor at higher concentration range up to 1,000 ppm (Figure 1 – Figure Supp 3) and characterized response to step functions (Figure 1 -Figure Supp 2). These measurements show that the array of metal oxide sensors is sensitive across a large range of odorant concentrations, significantly larger than the 0-200 ppm we use in behavior experiments.

Selectivity: We added measurements with other molecules commonly used in worm and larval chemotaxis studies (Figure 2 c,e), specifically ethyl butyrate and benzaldehyde, in addition to butanone and ethanol and compare the metal oxide sensors to readings with a PID for each.

Temporal resolution: We added new figures showing the odor sensor’s temporal response to step-wise modulated odor flow (Figure 1 – Figure Supp 3). This result indicates that the sensors act on a time scale that is on the order of ~10 seconds and is comparable to the downstream PID readout. This temporal response is sufficient for the slow changes we expect in our stable odor gradients.

line 204: 1-second temporal resolution. What is the evidence for this? Is this a sampling rate or does it take into account the dynamics of the sensor as shown in the Tariq paper?

We apologize for the confusion. We meant to refer to sampling rate, not the temporal response. We have rewritten as “*1 Hz sampling rate*.” As described in response to the previous comment, and in response to the requested Essential Revisions, we have added new measurements, analysis, and new language to address the temporal response of the sensors.

line 251: "remove two sensor bars and replace them with agar" I think it needs to be clearer upfront that the sensors have to be removed to do agar measurements.

We now show in Figure 6 and Figure 1 – Supp 6 that sensor bars do not strictly have to be removed, but that tracking is suboptimal when all sensor bars are in place. We now better emphasize the two “modes” of experimental measurements early on in the manuscript, in the introduction section, that do or do not require removing the sensor bars:

“The array of sensors can be used two ways: the full array can be used to measure the generated gradient throughout the extent of the chamber, or parts of the array can be used on the borders to monitor, during behavioral experiments, the boundary odor profile in the chamber. By varying flow rates and the sites of odor introduction, we show a variety of odor profiles can be generated and stabilized.”

line 371: butanone chemotaxis in *Drosophila* larvae. should cite the Jung and Bhandawat 2015 paper here that closely examines butanone-evoked navigation in adult flies and compares it to vinegar-evoked navigation.

Added. We thank the reviewer for suggesting this reference:

“Although butanone is not as commonly used as a stimulus with *Drosophila* as with *C. elegans*, butanone is known to be attractive to larval flies (Dubin et al. 1995, Dubin et al. 1998) and has been variously reported to be attractive (Park et al. 2002, Jung et al. 2015) and aversive (Israel et al. 2022, Lerner et al. 2020) to adult flies. ”

line 399: "This last feature…sets this method apart from previous approaches" It is not clear to me why the spectroscopic approach could not in principle be used during behavior.

To our knowledge this last feature has not been demonstrated with a spectroscopic approach. In principle it may be possible, but one would have to consider technical challenges such as: (1) design the chamber to be compatible with simultaneous optical access for both infrared imaging and imaging of the animals; (2) confirm that the spectroscopic measurements are not detected by the animal (thermally or visually), do not interfere with animal behavior, and do not optically conflict with the illumination needed to visualize the animals; and (3) be fast enough to meaningfully scan through the environment within the time course of a single animal experiment (Louis et al. 2008 reported 1 minute per scan at one cross section).

line 405: "In the future, such tuning curves may form the basis of investigations into neural mechanisms driving the sensorimotor transformations underlying navigation" Lots of this has been done! Seems weird to say this as a future thing and not cite the many many circuit cracking papers in worms and flies that have been pursued with other apparatus.

We apologize for the confusion. We were trying to point out that we have a new ability to measure tuning curves with respect to local odor gradients in more complex odor landscapes (not radially symmetric). To avoid confusion we have removed the offending text.

We now reference additional sensorimotor literature:

“Past work focusing on neural circuit mechanisms for odor navigation in worms have often relied on coarser readout such as the chemotaxis index or using the distance to odor source as a proxy for the inverse of odor concentration. However, it has been shown that the exact odor concentration alters the animal's behavioral response to odor (Yoshida et al. 2012, Levy et al. 2020). Quantifying tuning curves with respect to more precise odor measurements may better constrain the investigations into neural mechanisms driving the sensorimotor transformations underlying navigation algorithms (Clark et al. 2013, Liu et al. 2020).”

lines 438-440: "at the quasi-equilibrium conditions used in our experiments the odor concentration in agar is related to the airborne odor concentration directly above it up to a scalar that we predict to be constant across the agar." But is this actually true? This is quite far from the claim in the abstract that you can precisely measure the concentration at the location of the animal.

We now directly measure the odor concentration immediately opposite the surface of agar, as shown in new Figure 6 and new Figure 1 – Supp 6. As we discuss in response to comments from Reviewers #1 and #2 above, we show that the odor landscape inferred from measurements on the boundary is within some fractional error (10%) from the quasi-equilibrium profile without agar. In other words, the inference for odor concentration is supported both experimentally (direct measurement) and theoretically (numerical example and derivation in the Appendix).

As mentioned in response to Reviewer #2 Public Review #4, we have removed the offending word “precise.”

We are careful not to claim that this technique *measures* concentration ‘in’ agar. But in the Appendix we argue that it is reasonable to assume that the odor concentration that the worm likely experiences at the gas-liquid interface is a scalar multiple of the odor concentration in air. It is also mentioned in the Discussion section:

“Our reaction-convection-diffusion model suggests that at the quasi-equilibrium conditions used in our experiments the odor concentration in agar is related to the airborne odor concentration directly above it up to a scalar that we predict to be constant across the agar.”

Reviewer #3 (Recommendations for the authors):In this paper, Chen et al. propose a new method to measure odor stimuli in space and time. Measuring the odor stimulus is a key step in interpreting odor-driven behavior and understanding the neural mechanisms that mediate it, but this task still challenges every experimenter in the field. As described in the introduction, there is basically not a single method that is good for most behavioral assays, even when these involve small animals such as *C. elegans* and *D. melanogaster*. Previous approaches are either invasive, very expensive, or limited to very small behavioral arenas. The strength of the proposed method is to be cheap and to have a reasonable spatial (1 cm) and temporal (1 s) resolution. The full sensor array introduced here cannot be located throughout the behavioral arena, but a mathematical model shows that it is sufficient to measure the odor stimulus at certain specific positions to reconstruct the full spatial profile. In other assays, such mathematical considerations might not be possible and physical constraints might make it impossible to use the sensor during behavior, however, it can still be used to measure the odor landscape in a separate experiment, and, with a certain degree of reproducibility (which will depend on the specific delivery system), this is still better than no measure. It should be noted that the method is limited to 2D measurements, which is sufficient for the walking or crawling behavior of small animals in non-turbulent conditions, but it cannot be extended to 3D assays. It remains moreover unclear what is the sensitivity range of the sensor for the odors used in the paper (butanone and ethanol) and what is expected for other odorants (which compounds are detectable?).

We thank the reviewer for positive comments and suggestions. We now clarify in the text that the method is restricted to studying crawling behavior in 2D environments:

“Future work is needed to improve concurrent animal tracking methods, to design more miniaturized and scalable sensors, and to generalize to more complex or three-dimensional environments.”

We clarify a potential misunderstanding: in all experiments with agar we measure odor sensors concurrent with behavior, but these measurements typically occur on the boundary around the agar. The exception is Figure 6 and Figure 1 – Supp 6 which measure directly opposite the surface of the agar, including when animals are present (Figure 1 – Figure Supp 6). We show how this creates difficulty for tracking animal behavior, which motivates our preferred approach to measure along the boundary during behavior.

In new experiments we further characterize the sensitivity range of the sensors across a 0-1000 ppm range (Figure 1 – Figure Supp 3). And we conduct new experiments to demonstrate use with oderants benzaldehyde and ethyl butyrate (Figure 2 c and e) in addition to the butanone and ethanol shown previously. We describe this in more detail in our response to requested Essential Revisions.

As proof of principle, the authors use this new method to characterize the behavior of *C. elegans* and *D. melanogaster* as a function of the concentration gradient encountered along their moving trajectories. In this respect, it remains unclear whether the method allows new conclusions or simply recapitulates previous observations.

Some of our conclusions are new: for example, while the parallel use of both weathervaning and biased random walk strategies had been previously well characterized for other forms of chemotaxis, such as salt chemotaxis (Iino et al. 2009, Luo et al. 2014), to our knowledge we are the first to characterize navigation in airborne butanone. We now state:

“We conclude that *C. elegans* utilize both biased random walk and weathervaning strategies to navigate butanone airborne odor landscapes (Iino et al. 2009, Yoshida et al. 2012). Both strategies had been observed for salt chemotaxis before, but to our knowledge had not previously been quantitatively characterized for airborne butanone. More broadly, previous measurements of navigation in response to airborne odors for *C. elegans* have not had access to the underlying odor concentration and therefore could only use proxies, such as the bearing angle or distance with respect to a point odor source. Our approach allows a more direct description of the navigation strategy because we characterize turn probabilities and other features with respect to the local gradient directly (FIGURE 7), as opposed to a proxy.”

We also added new experiments that for the first time characterize the changes to butanone navigational strategy induced by defects in interneuron AIB. AIB’s role in weathervaning had previously been studied in the context of salt chemotaxis, but not in the context of butanone olfactory guided navigation. We now clarify:

"To identify potential neurons involved in mediating the navigation strategy, we measured butanone guided navigation of mutant animals in which the interneuron AIB was chronically inhibited. Disruptions to AIB had previously been shown to either decrease weathervaning (AIB ablation) (Iino et al. 2009) or decrease biased random walks (Luo et al. 2014) (AIB inhibition) when navigating in a salt gradient. But to our knowledge, AIB's role in influencing the animal's butanone-guided navigation strategy had not previously been measured. We observed that AIB-inhibited animals still climbed butanone gradients, but the worm relied on different behavioral strategies (FIGURE 8). The turn was still biased suggesting that the animal still used a biased random-walk strategy, although turning rate was overall lower. The animal showed down-regulated weathervaning. In wild-type (N2) animals the distribution of the curvature of the animal's trajectory depends on its orientation to the local odor gradient (FIGURE 7f, Figure 7 -Figure Supp 2a) indicating weathervaning, but in AIB-inhibited animals the distribution of trajectory curvature is much less dependant on orientation to the local gradient (FIGURE 8d, Figure 7 – Figure Supp 2b). Together this suggests that AIB is important for supporting turning rate and the weathervaning strategy, and demonstrates the potential for this approach to characterize mutants or investigate neural circuits underlying navigation."

We agree with the reviewer that an important aspect of this work has been to recapitulate previous findings using our new approach. This is intentional and consistent with its submission as a “Tools and Resources” article, which explicitly “*do not* have to report major new biological insights or mechanisms.*”* We address this further in response to the 6th point in the requested Essential Revisions.

Sensors:It would be important to know more about the sensitivity range of the sensors. It seems that the concentration used here does not saturate the sensors: what is the full scale of sensitivity for the x-axis in Figure 1- S1? Is the calibration curve similar for ethanol? And what kinds of odors are expected to be detectable?Another point that is not mentioned is whether the sensor itself affects the odor stimulus: for example, through an absorption/release mechanism similar to what happens with the agar: I guess one should compare Figure 2a to Figure 4b? No interference with the stimulus would be a clear advantage of this approach over the PID and should be stated.

As described in response to requested “Essential Revisions'', we now included new measurements to characterize the metal-oxide based odor sensors across a range of odor concentrations up to 1,000 ppm (Figure 1 – Figure Supp 3). We also characterize their temporal response profile and their stability (Figure 1 – Figure Supp 2). We now perform new measurements to compare the metal-oxide sensors to PIDs separately for four odorants that span different water solubilities and vapor pressures (Figure 2 – Figure Supp 1): butanone, benzaldehyde, ethyl butyrate, and ethanol, as described in response to the third requested Essential Revision. We now clarify:

“To our knowledge, most volatile organic compounds that can be sensed and calibrated by a PID should be compatible with our sensor array system.”

In new measurements shown in Figure 1 – Figure Supp 3b we observe that the downstream odor concentration measured by the PID is not affected by powering on or off the metal oxide odor sensors. This suggests that the metal oxide sensors are not depleting the odorant during measurements. We agree that metal oxide sensor’s lack of interference with the stimulus is an advantage over the PID.

Behavioral analysis:Regarding the paragraphs on *C. elegans* and *D. melanogaster*, I would suggest that the authors clarify what is a new finding vs what is already known, and in which cases the measurement of the odor gradient is critical.Clearly, Figure 7 requires such measurements; however, the significance of the result is somehow obscure: what do we expect for the relationship between drift velocity and gradient? Also, there seems to be a very small and possibly not significant (I do not find a statistical test) positive drift for the high range of the tested gradients: could these gradients be too shallow?

We now clarify what is a new finding and what has been known for worm chemotaxis:

“We conclude that *C. elegans* utilize both biased random walk and weathervaning strategies to navigate butanone airborne odor landscapes (Iino et al. 2009, Yoshida et al. 2012). Both strategies had been observed for salt chemotaxis before, but to our knowledge had not previously been quantitatively characterized for airborne butanone. More broadly, previous measurements of navigation in response to airborne odors for *C. elegans* have not had access to the underlying odor concentration and therefore could only use proxies, such as the bearing angle or distance with respect to a point odor source. Our approach allows a more direct description of the navigation strategy because we characterize turn probabilities and other features with respect to the local gradient directly (FIG7), as opposed to a proxy.”

We have also added a new experiment studying navigational strategy of an AIB defective mutant resulting in another new finding:

“To identify potential neurons involved in mediating the navigation strategy, we measured butanone guided navigation of mutant animals in which the interneuron AIB was chronically inhibited. Disruptions to AIB had previously been shown to either decrease weathervaning (AIB ablation) (Iino et al. 2009) or decrease biased random walks (Luo et al. 2014) (AIB inhibition) when navigating in a salt gradient. But to our knowledge, AIB's role in influencing the animal's butanone-guided navigation strategy had not previously been measured….”

We have added more statistical rigor in figure 9 (the original figure 7). We now perform t-tests for each bin against the null-hypothesis (zero drift velocity) and found significant differences for every bin other than the first. Given the non-Gaussian distribution, we also performed two-sample Kolmogorov-Smirnov between bins and found that they are all significantly different from the lowest drift velocity bin. These statistical tests are incorporated in the revised figure 9 caption:

“Other than the left-most bin with the lowest gradient, all other bins show significant difference from zero drift velocity using t-test (p<0.001, ***). Both two-sample t-test and Kolmogorov-Smirnov test (given the non-Gaussian distribution) show significant differences between bins and the first bin with the lowest concentration gradient (p<0.001).”

We do not find the amplitude of the drift velocity to be shallow compared to our expectations of common worm chemotaxis kinetics: The average arrival time from the center of the plate to droplet source 5 cm away is normally on the order of ~10 minutes. Our measured mean drift velocity reported in Figure 9 is ~0.1 mm/s which corresponds to ~ 8 minutes and is roughly consistent with what we would expect. We think that this measurement has a wide spread due to the stochastic behavior in worm chemotaxis, and can be compared to similar saturating drift velocity in bacterial chemotaxis (Mattingly et al. 2021). It is interesting to consider whether the drift velocity tuning to local gradients might be dependent on the gradient distribution of the environment. Investigating how behavioral performance changes as a function of the gradient statistics would be an interesting future direction.

Is the same analysis of drift vs gradient not possible with *D. melanogaster*? Moreover, I wonder why the turn rate for -90 and 90 degrees do not have similar values: both directions are perpendicular to the gradient, wouldn't one expect the same behavior? The heading change, in that concern, is as expected, with similar absolute value and opposite direction. Also unclear why the heading change is similar for 180 and 0 degrees.

We hesitate to apply the drift vs gradient analysis to *Drosophila* larvae because their runs have a much longer persistence length than *C. elegans* which could confound the interpretation given the relative small size of our arena. We chose to optimize the arena size for *C. elegans*. We added *Drosophila* experiments later to demonstrate compatibility. If one was interested in a dedicated instrument for *Drosophila,* one could consider fabricating a larger arena (~30cm, as we used previously in Gershow et al., 2012) with identical design principles and larger odor sensor bars.

We also expected the turn rates of *Drosophila* larvae to be the same for -90 and 90 degrees; that they differ likely reflects the limited size of the data set used in this analysis. The average heading change is signed, so a non-zero average represents a directional bias (e.g. positive average heading change represents a bias towards the counter-clockwise or left-hand direction). We do not expect a directional bias for the 0 or 180 degree directions, and therefore expect both to be nearly 0, in agreement with the data presented in Figure 10b.